# Effect of prolonged sitting immobility on shear wave velocity of the lower leg muscles in healthy adults: A proof-of-concept study

Kumiko Okino[1]◉, Mitsuhiro Aoki◉[2]*, Masahiro Yamane[3]◉, Chikashi Kohmura[1]◉

**1** Department of Clinical Laboratory Medicine, School of Medical Technology, Health Sciences University of Hokkaido, Kita-ku, Sapporo, Japan, **2** Department of Physical Therapy, Graduate School of Rehabilitation Science, Health Sciences University of Hokkaido, Tobetsu-cho, Ishikari-gun, Japan, **3** Department of Physical Therapy, Health Science University Hospital, Kita-ku, Sapporo, Japan

◉ These authors contributed equally to this work.
* aokimotionpicture2008@gmail.com

## Abstract

### Objective

The purpose of this study is to investigate the physical changes of the lower leg muscles in the compartment by observing the changes in the shear wave velocity of the gastrocnemius, soleus and tibialis anterior muscles with time in the sitting position for 2 hours and after elevation of the lower leg.

### Materials and methods

The subjects were 24 healthy adult males (average age 26.6 years). Shear wave velocity was measured by Aplio 500 in immobilized leg immediately after the start of sitting, 60 minutes and 120 minutes after the start of sitting. After 120 minutes the subjects raised the lower leg for 3 minutes, then measured again.

### Results

In the lateral and medial gastrocnemius, there was a significant increase in the velocity at 60 (1.58 ± 0.06, 1.70 ± 0.09 m/s) and 120 minutes (1.70 ± 0.10, 1.83 ± 0.11 m/s) after the start of the test (1.52 ± 0.06, 1.66 ± 0.10 m/s), respectively (p<0.01). In the soleus and the tibialis anterior, there was a significant increase in the velocity at 120 minutes (1.89 ± 0.17, 2.30 ± 0.24 m/s) compared to after the start (1.60 ± 0.15, 2.15 ± 0.26 m/s), respectively (p<0.01). In all muscles, there was a significant decrease in the velocity after the raising compared to that of 120 minutes (p<0.01).

### Conclusions

It has been reported that the change of shear wave velocity with time is proportional to the intramuscular pressure in the leg compartment, and it is assumed that the increase of shear wave velocity in the 2-hour seated leg is due to fluid retention in extra-cellular space of the compartment.

**Data Availability Statement:** All relevant data are within the paper and its Supporting Information file.

**Funding:** The authors received no specific funding for this work.

**Competing interests:** The authors have declared that no competing interests exist.

## Introduction

In recent years, advances in shear wave elastography have made it possible to evaluate the mechanical properties (elastic modulus) of skeletal muscles in real time by measuring shear wave velocity. It is possible to measure the conditions of stretched muscles (static mechanical properties) and that of active muscle contraction (dynamic mechanical properties) [1, 2].

The measurement of elastic modulus of skeletal muscle began around 2005, and a series of reports by Gennisson et al. (2005, 2010) established that the elastic modulus of muscle can be accurately measured by placing a probe parallel to the muscle fibers [1, 3]. This has been confirmed by subsequent studies, which have shown that shear modulus measurements reflect the physical properties associated with muscle extension and contraction. At the same time, they evaluated the difference between anisotropy and isotropy of ultra-sound transmission brought about by transverse and longitudinal probe placements. Longitudinal measurements have become the most common way to measure the elastic modulus of muscle tissue in recent years [4].

Elastic modulus measurements of the triceps surae muscle in resting condition were initiated by Lacourpaille et al. (2012) and Maisetti (2012) [2, 5]. An attempt to estimate the resting muscle tension during passive motion of the ankle and knee joints was made mainly in the medial and lateral gastrocnemius and soleus muscles, where the slack angle was determined. The elastic modulus was increased by the muscle elongation, in which the muscle was stretched from the slack length [6–8].

Regarding the difference between elastic modulus and shear wave velocity, Eby (2013) found that the equation $E = 3\rho Vs^2$ (E: elastic modulus kPa, Vs: shear wave velocity m/s, $\rho$: muscle density 1000 kg/m$^3$) holds if the density of the muscle is assumed to be uniform, in case of measurement along longitudinal muscle fiber direction [4]. On the contrary, if the density of muscles is not uniform, in case of measurement along transverse or oblique muscle fiber direction, it is expressed in terms of the anisotropic modulus ($E = \rho Vs^2$) [2, 7]. In researches of the shear wave velocity in the leg muscles at a resting state (ankle position around the slack angle without muscle contraction), the muscle tissue was measured regarding as the isotropic modulus ($E = 3\rho Vs^2$) [2, 5, 7–9].

The muscles of the lower leg are divided into four compartments: the anterior compartment consists of the tibialis anterior (TA) and the extensor hallucis longus and extensor digitorum longus, the anterior-lateral compartment consists of the peroneus longus and brevis, the superficial posterior compartment consists of the triceps surae muscle (the lateral and medial gastrocnemius (LG and MG), and soleus (SOL)), and the deep posterior compartment consists of the tibialis posterior and flexor hallucis longus and flexor digitorum longus. Compartment syndrome occurs when the internal pressure in each compartment increases due to excessive sports activity or severe trauma [10]. In addition, the gastrocnemius and soleus muscles in the posterior compartment contain large sinusoid veins, and venous thrombosis is known to occur when blood flow is stagnant [11].

There are several clinical conditions that cause lower leg symptom during a prolonged sitting, i.e. occupational leg edema after prolonged standing and sitting [12], thrombophlebitis of lower leg during long flight [13, 14] and fluid accumulation in the leg during long-distance bus travel [15]. To solve mechanism of prolonged sitting in lower leg, plethysmographic measurement for haemodynamics of calf veins or bioelectrical impedance analysis for tissue fluid has been performed [16, 17], but no measurement of resting leg muscles by shear wave elastography has been reported.

The purpose of this report is to measure the shear wave velocity of the tibialis anterior, gastrocnemius and soleus muscles by sitting for 2 hours and then raising the leg, and to clarify the physical changes of the leg muscles in the compartment.

We hypothesized that the shear wave velocity of the medial and lateral gastrocnemius, soleus and tibialis anterior muscle would increase with time in the resting leg position after 2 hours of sitting and decrease with subsequent leg elevation. Then we considered the mechanism of the velocity change with two points of view, i.e. one is change of leg muscle fibers and the other is change of extra-cellular space in the compartment.

## Materials and methods

### Subjects

Twenty-four healthy adult males (mean age, 26.6 ± 6.1 years) were included in the study. Subjects were recruited by announcements on posters which were exhibited in a student bulletin board. Subjects with blood disorders or traumatic diseases of the lower legs were excluded in advance. The subject's height (172.4±7.4 cm), weight (65.8±7.8 kg), and BMI (22.0±2.5) were measured, and lower extremity muscle mass (9.8±1.0 kg), body fat mass (14.4±9.3 kg), and trunk muscle mass (26.0±2.5 kg) were measured using a Tanita body composition analyzer (MC-180, Tanita Ltd., Tokyo).

The purpose of the study, management of personal information, prohibition of using personal information for other purposes, and free participation in the study were explained to the subjects and their written consent was obtained. The individual in this manuscript has given written informed consent to publish obtained data including images of the participants. In accordance with the Declaration of Helsinki, the subject's personal right and data protection was explained. This study was conducted with the approval of the University of Health Sciences Research Ethics Review Committee (approval number, 19R115109).

### Experimental protocol

Subjects were seated in a reclining chair and held in a relaxed position for 2 hours. Shear wave velocity of the soleus, medial and lateral gastrocnemius, and tibialis anterior muscles of the left lower extremity were measured immediately after the start of a sitting, 60 minutes and 120 minutes later using an ultrasound system (Aplio 500 Canon, Tokyo). Afterwards, the patient raised left lower limb for 3 minutes and returned to the sitting position for measurement (Fig 1). Measurements were performed by a clinical laboratory technician (K.O.). The temperature and humidity in the laboratory were maintained at 25˚C and 30–40%, respectively.

Shear wave velocity was measured after start of sitting on a reclining chair, at 60 min, and at 120 min. After 3 min leg raise on the stool, the velocity was measured again in sitting position.

**Sitting posture.**   The subject was seated in a reclining chair and maintained 20 degrees of flexion at the ankle joint, 70 degrees of flexion at the knee joint, and 80 degrees of flexion at the hip joint as measured by a goniometer. In this limb position, the ankle is more flexed than the slack angle, which relaxes the triceps surae muscle [6–8]. In this limb position, the ankle joint was flexed for the tibialis anterior muscle to keep slightly elongated. The subjects were instructed to sit in a seated position with the width of the right and left legs shoulder-width apart and to maintain a relaxed state during the measurement, and the shear wave velocity was measured. The foot was held on the weight-bearing platform so that the foot load was equally applied to both soles (Fig 2A).

**Elevation of the lower limb.**   After 120 minutes of measurement, subjects were seated in a reclining chair with the knee joint extended and the lower leg raised on a stool at the same height as the thigh for 180 seconds to relax the lower leg and ankle, and then returned to the sitting position to measure the shear wave velocity (m/s) (Fig 2B). Participants remained in the same position sitting in a chair from the beginning until the end of the elastography measurement, except 3 minutes leg elevation.

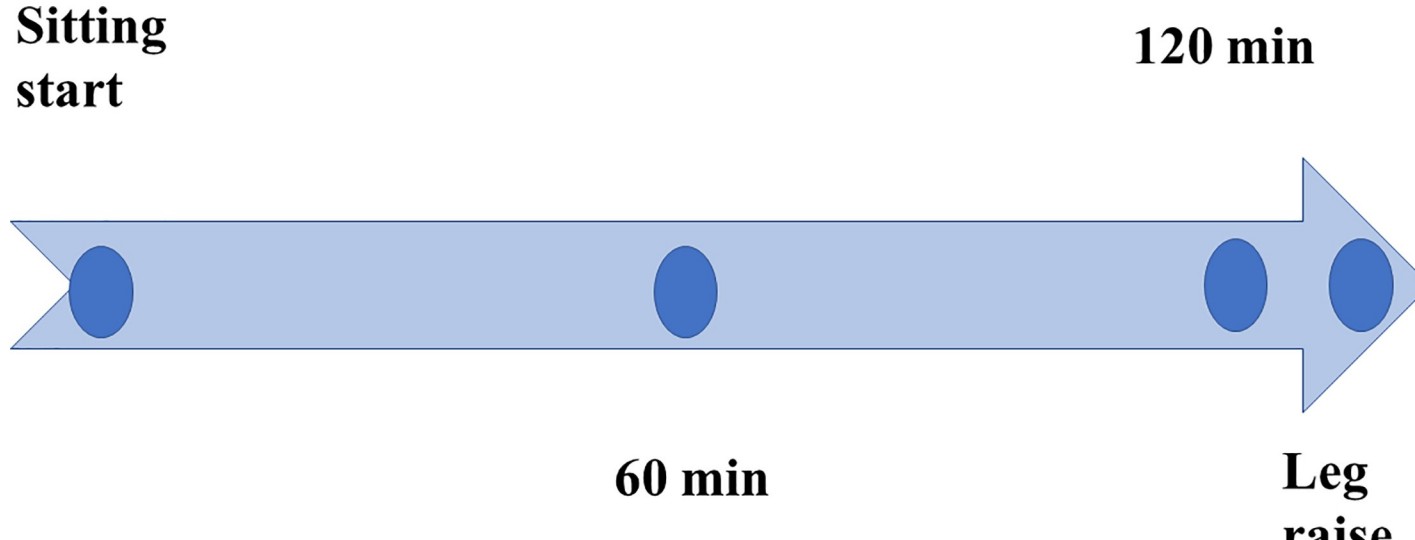

**Fig 1. Experimental design.**

**Measurement of shear wave velocity.** Shear wave velocity was measured as an indicator of muscle stiffness. We used an ultrasound system (Aplio 500, Canon, Tokyo) and a linear probe (PLT100BT5, Canon, Tokyo) with a field view of approximately 58 mm, standard operating frequency of 10 MHz (up to 14 MHz). The accuracy of the velocity measurement by Aplio500 using a phantom (CIRS, Norfork, Virginia, USA; model 049 and 049A) of 8.0 ± 3.0 kPa is 0.49 kPa, and the precision (Coefficient of Variation and Confidence Interval) is 6.96%, 5.79–8.13%. The accuracy and precision of the velocity measurements from 2 to 6 cm material depth are not different [18]. The range of measurement of the shear wave velocity of the leg muscles in this study by Aplio500 was 1.0–3.0 m/s (3.0–9.0 kPa), and the depth of the soleus muscle at the deepest point was about 4.0 cm, so the reliability of the measurement of the shear wave velocity of the leg muscles by Aplio 500 was assured.

The measured muscles were the medial and lateral gastrocnemius, soleus and tibialis anterior. The location of a probe to be applied on the muscle belly of the medial and lateral gastrocnemius, soleus and tibialis anterior muscles was adjusted in the designated position (the medial and lateral portions of the gastrocnemius muscle at 30% of the proximal left lower leg,

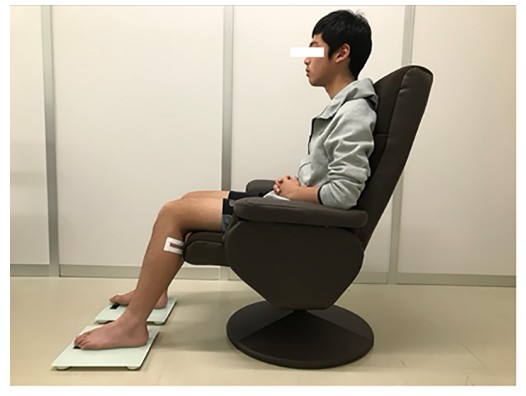 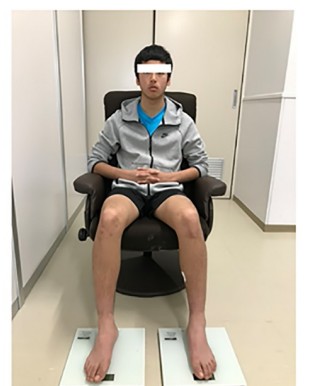 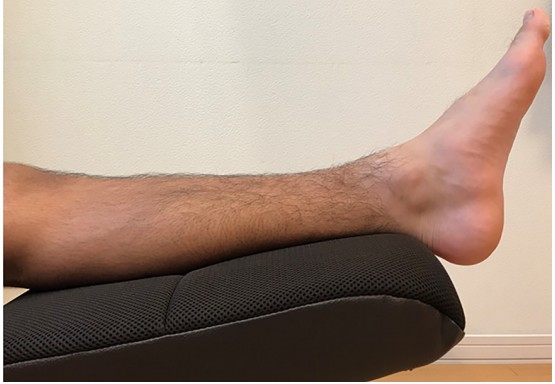

**Fig 2.** a. Sitting position on a reclining chair. b. Leg raise on a stool.

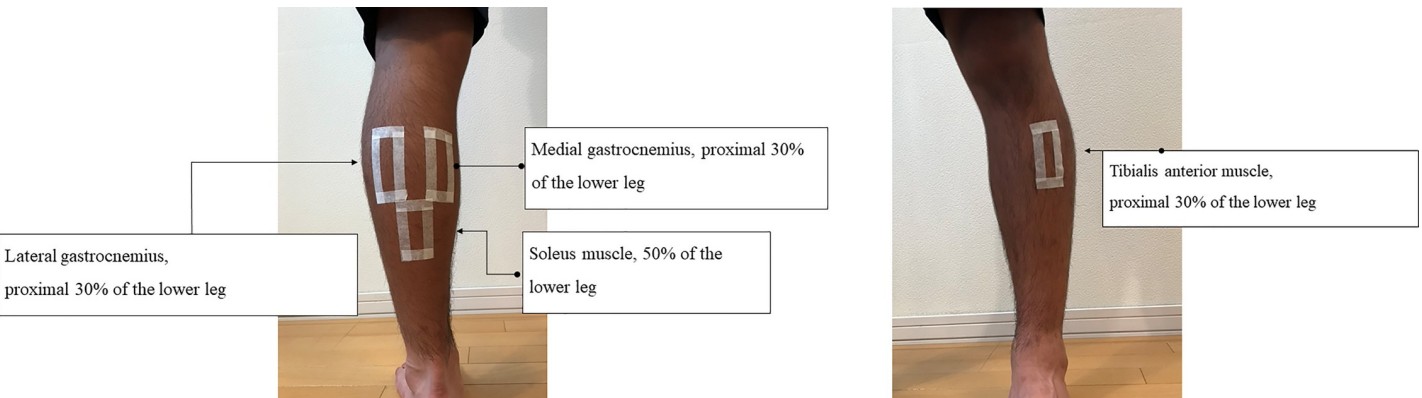

**Fig 3.** a. Sites of recording of shear wave velocity at lateral, medial gastrocnemius, and soleus muscles. b. Sites of recording of shear wave velocity at tibialis anterior muscle.

the soleus muscle at 50% of the left lower leg, and the tibialis anterior muscle at 30% of the proximal left lower leg [2, 5]) for individual subject. On the most prominent part of the muscle belly, the rectangular shape skin mark with adhesive tape was attached prior to the measurement, and the probe was placed in the rectangular mark (Fig 3A and 3B). Since orientation of muscle fascicles of the gastrocnemius, soleus is oblique by their pennated structure and is redundant due to muscle relaxation, prove setting was aligned to anatomical muscle shortening direction.

Ultrasound images were recorded along the long axis of each muscle. The probes were scanned parallel to the superficial fascia of each muscle. The size of the Region of Interest (ROI) was set to 10×10 mm$^2$. The area of analysis (Target ROI) was set as a circle of 5 mm in diameter in the center of the ROI [19]. The ultrasound images were taken three times for each muscle and the average of the three images obtained was calculated (Fig 4).

The shear wave velocity of the gastrocnemius and soleus muscles was measured at a value with the ankle joint in 20 degrees of flexion and muscle length was shorter than the slack length [20]. The shear wave velocity of the tibialis anterior muscle was measured with the ankle joint in 20 degrees of flexion at 10 degrees over the slack angle and with the muscle length slightly longer than the slack length [21]. Taking both longitudinal measurements of the muscle and muscle length into consideration, in the present measurements, we regarded both muscles as isotropic and elastic modulus was calculated by the following equation [7].

$$E = 3\rho Vs^2$$

**Surface electromyography.** In order to confirm the contraction of the leg muscles in the reclining seat, electromyography was measured by placing active wireless surface electrodes (Nihon Koden Ltd., Tokyo, Japan) with a distance of 10 mm between electrodes parallel to the muscle fibers in the medial and lateral gastrocnemius, soleus muscle and tibialis anterior muscle. Surface electrode placement sites were adjusted according to Hermens et al. (2000) [22]. The electromyography data was recorded at 2000 Hz and collected to a polygraph system of Web-1000 (Nihon Koden Ltd., Tokyo, Japan), then analyzed using LabChart version 7 (AD Instruments, Tokyo, Japan). During shear wave velocity measurement by ultra-sonography, EMG data was continuously displayed for examiner and subject on PC screen, suggesting the meaning of appearance of muscle activation waves for ensuring proper relaxation. Appearance of EMG activity was determined by observation of action potentials greater than 2SD amplitude of baseline. It was determined that no muscle contraction was observed to occur in our lower leg sitting position.

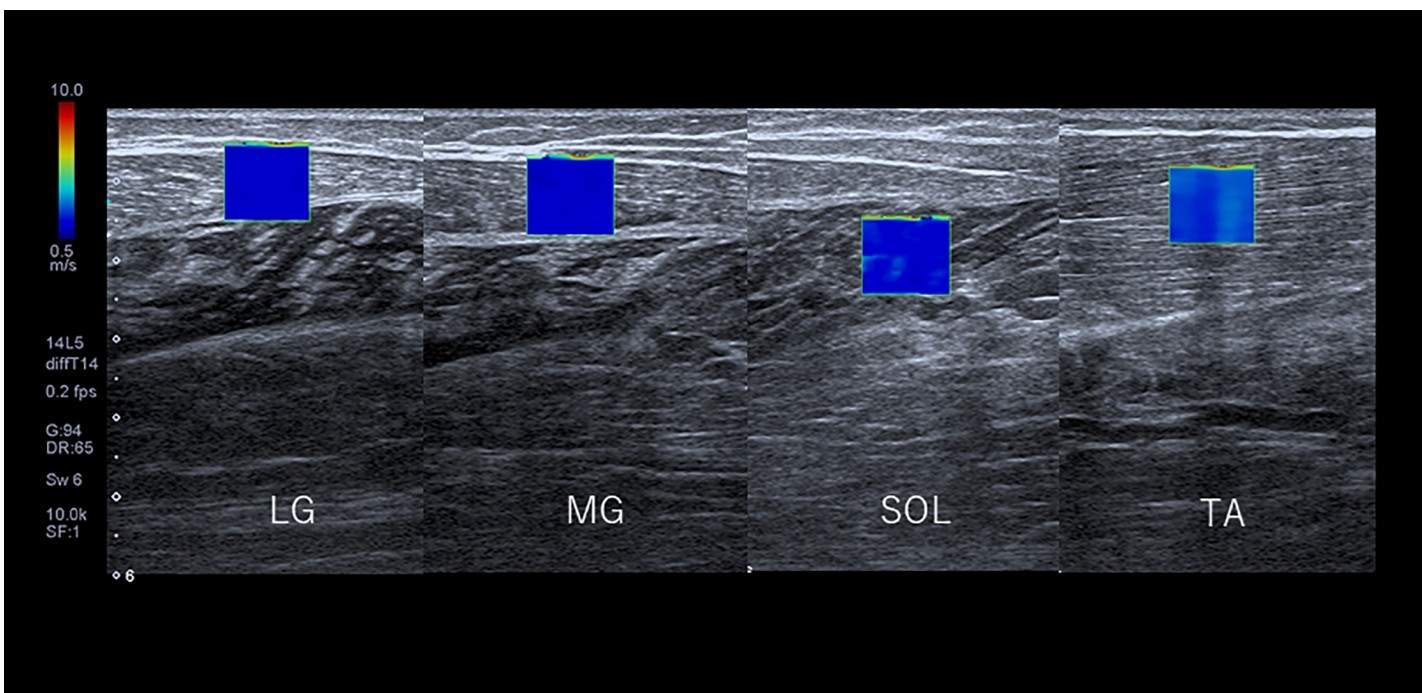

**Fig 4. Image of shear wave elastography of lower leg muscles.** LG: Lateral gastrocnemius, MG: Medial gastrocnemius, SOL: Soleus, TA: Tibialis anterior. ROI: Region of interest: 1 cm x 1 cm.

**Reliability analysis.** In order to evaluate the intra-observer reliability and accuracy of shear wave velocity measurements, shear wave velocity was measured in the medial and lateral gastrocnemius, soleus and tibialis anterior muscles in reclining seated sitting position in eight healthy male adults. Measurements were taken immediately after sitting and 20 minutes later, under the same conditions as in the present experiment, to obtain the intra-day intra-observer reliability (1, 3) and the coefficient of variation (CV). Intra-day intra-observer reliability and coefficient of variation (CV) for the lateral gastrocnemius muscle was 0.88 and 3.9%, for the medial gastrocnemius, 0.87 and 5.7%, for the soleus muscle, 0.92 and 7.5%, and for the tibialis anterior muscle, 0.89 and 14.2%. After one more day, measurements were performed again, the inter-day intra-observer reliability (1, 3) and the coefficient of variation (CV) was obtained. The inter-day intra-observer reliability and the correlation coefficient (CV) for the lateral gastrocnemius was 0.77 and 3.7%, the medial gastrocnemius was 0.82 and 6.5%, the soleus was 0.76 and 8.7%, and the tibialis anterior was 0.81 and 12.0%.

**Sample size.** To determine the sample size for repeated measures two-way analysis of variance, we used Cohen's moderate effect size of 0.25, calculated the group as 2, the level as 4, the α value as 0.05 and the β value as 0.8, and set the sample size as 24 [23]. Therefore, the number of subjects in this study was set as 24.

## Statistical analysis

Information of subjects were analyzed by the Shapiro-Wilk test, and the normality of all values was confirmed. The measured shear wave velocities from four different muscles was analyzed by the Shapiro-Wilk test, and the normality of all the values was confirmed. Measurements obtained with time were analyzed by repeated-measures two-way analysis of variance, and no interaction was found for the combination of LG and TA or MG and TA. However, because

there were interactions between SOL and TA, and between LG, MG, and SOL with each other, repeated-measures one-way analysis of variance was performed for individual muscles. The Bonferroni method was used for the post-hoc test. A corresponding t-test was used for inter-muscle comparisons of shear wave velocities of immediately after the start of a sitting. Statistical analysis was performed using SPSS ver23, and level of significance was set at 0.05.

## Result

### Change in the shear wave velocity of the leg muscles with time

In the lateral and medial gastrocnemius muscles, the shear wave velocity increased significantly at 60 minutes ($1.58 \pm 0.06$, $1.70 \pm 0.09$ m/s) and 120 minutes ($1.70 \pm 0.10$, $1.83 \pm 0.11$ m/s) compared to immediately after the start of sitting, respectively ($1.52 \pm 0.06$, $1.66 \pm 0.10$ m/s) ($p < 0.01$, $p < 0.05$). And the velocity after leg raising ($1.52 \pm 0.07$, $1.59 \pm 0.07$ m/s) is significantly lower than the velocity at 120 minutes, respectively ($p < 0.01$, $p < 0.01$) (Table 1, Figs 5 and 6).

In the soleus and tibialis anterior muscles, the shear wave velocity increased significantly after 120 minutes ($1.89 \pm 0.17$, $2.30 \pm 0.24$ m/s) compared to immediately after the start of sitting ($1.60 \pm 0.15$, $2.15 \pm 0.26$ m/s), respectively ($p < 0.01$, $p < 0.05$). And the velocity after leg raising ($1.55 \pm 0.11$, $2.03 \pm 0.22$ m/s) is significantly lower than the velocity at 120 minutes, respectively ($p < 0.01$, $p < 0.01$). (Table 1, Figs 7 and 8).

**Comparison of shear wave velocity between muscles immediately after the start of sitting.** The shear wave velocity of the tibialis anterior muscle was significantly greater than that of the other muscles ($p < 0.01$) (Table 1). There was no difference in the shear wave velocity among the medial and lateral gastrocnemius and soleus muscles, except the shear wave velocity between the lateral gastrocnemius and medial gastrocnemius muscle ($p < 0.01$).

## Discussion

Taniguchi (2015) reported that in the lateral gastrocnemius and medial gastrocnemius muscles stretched beyond the slack length, the elastic modulus was unchanged (7.5 kPa in the lateral gastrocnemius and 10.5 kPa in the medial gastrocnemius) after 20 minutes of resting in the standing position [6]. There is no other report on the measurement of the shear wave velocity in the leg muscles under the prolonged resting condition.

In this study, we observed the shear wave velocity of the leg muscles in sitting position and its change with time. As the gastrocnemius and soleus muscle located behind the lower leg were measured in a relaxed position, their length is shorter than the slack length. As the tibialis

**Table 1. Shear wave velocity of LG, MG, SOL, and TA over time.**

| | Start of sitting | 60 min | 120 min | After leg raise |
|---|---|---|---|---|
| LG | $1.52 \pm 0.06$ | $1.58 \pm 0.06^*$ | $1.70 \pm 0.10^*$ | $1.52 \pm 0.07^{**}$ |
| MG | $1.66 \pm 0.10$ | $1.70 \pm 0.09^*$ | $1.83 \pm 0.11^*$ | $1.59 \pm 0.07^*$ |
| SOL | $1.60 \pm 0.15$ | $1.69 \pm 0.15$ | $1.89 \pm 0.17^*$ | $1.55 \pm 0.11^{**}$ |
| TA | $2.15 \pm 0.26$ | $2.21 \pm 0.24$ | $2.30 \pm 0.24^*$ | $2.03 \pm 0.22^{**}$ |

Unit: m/s.

Average ± SD.

LG: Lateral gastrocnemius, MG: Medial gastrocnemius, SOL: Soleus, TA: Tibialis anterior

$^*$p<0.05, Shear wave velocity is greater than that of start of sitting.

$^{**}$p<0.01, Shear wave velocity of after leg raise is smaller than that of 120 min.

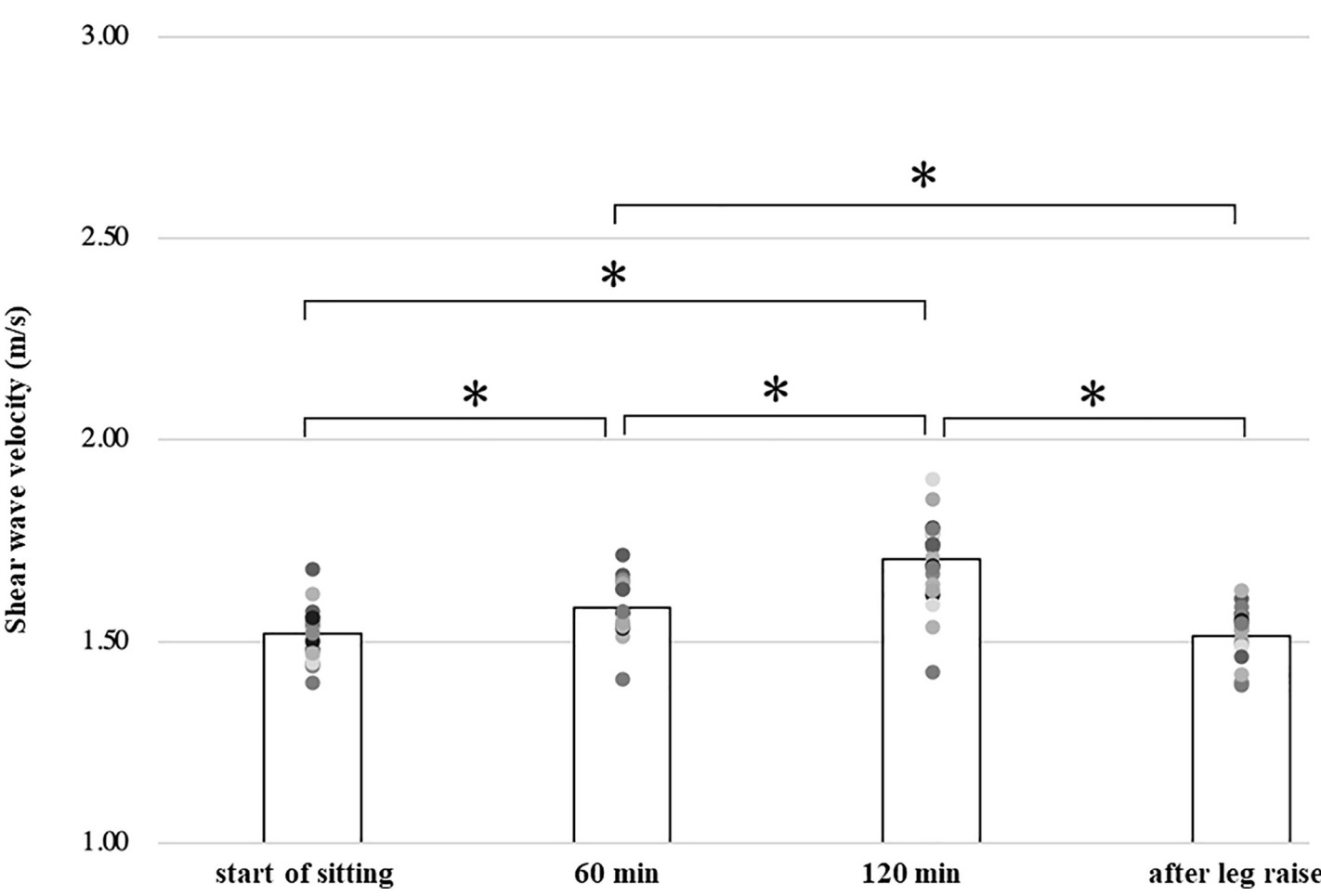

**Fig 5. Shear wave velocity of the lateral gastrocnemius.**

anterior muscle located anterior to the lower leg was measured at the ankle flexed position, its length is longer than the slack length. Shear wave velocity in the lateral gastrocnemius, medial gastrocnemius, and soleus muscles increased significantly during 120 minutes of sitting compared to the immediately after sitting and decreased significantly during 3 min leg raising (Table 1, Figs 5–7). The shear wave velocity in the tibialis anterior muscle was significantly greater than those in the gastrocnemius and soleus muscles (p<0.01). It increased significantly with time, and decreased significantly after elevation of the lower leg (Table 1, Fig 8).

Toyoshima (2020) reported using Turkey legs that the shear wave velocity increased proportionally with the increase in pressure of the tibialis anterior compartment [24]. Specifically, when the internal pressure of the tibialis anterior muscle is 1–2 mmHg, the shear wave velocity (elastic modulus) is 2.5 m/s (6.25 kPa) and increases linearly from 3.0 m/s (9.0 kPa) to 3.5 m/s (12.5 kPa) as the internal pressure increases from 10 mmHg to 20 mmHg. In the reports of human measurement, shear wave velocity (elastic modulus) of the tibialis anterior muscle at rest has been reported to be 2.1–3.2 m/s (4.4–10.2 kPa) [4, 9, 22, 23]. In our measurements, the shear wave velocity (elastic modulus) of the tibialis anterior muscle was 2.17 m/s (6.51 kPa), which is similar to the previous reports (Table 1, Fig 8).

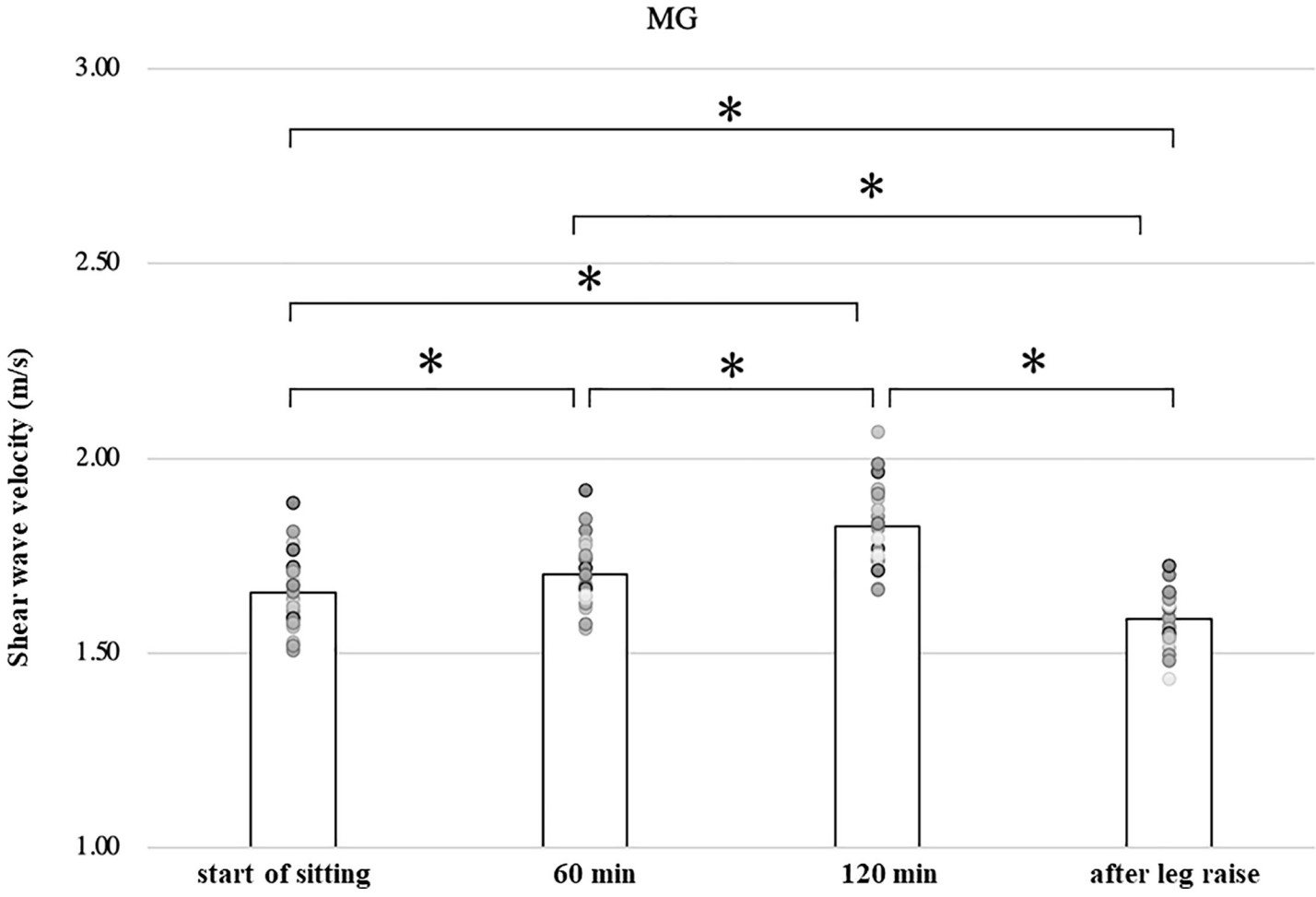

**Fig 6. Shear wave velocity of the medial gastrocnemius.**

In sitting position, venous pressure of popliteal vein at the popliteal fossa and pressure of saphenous vein at the medial malleolus were reported to be 23.9 mmHg (3.17 kPa) and 54.2 mmHg (7.2 kPa), respectively [25]. Considering the location of shear wave velocity measurements in the tibialis anterior and gastrocnemius muscles in the sitting position, it can be inferred that hydrostatic pressure is equally applied to the tibialis anterior and gastrocnemius muscles through the leg veins for two hours.

In the present measurement, the shear wave velocity of the tibialis anterior muscle was significantly greater than that of the gastrocnemius and soleus muscles at any times (Table 1). This is because the tibialis anterior muscle is stretched about 10% longer than the slack length at the sitting leg position, which may be one of the reasons why the shear wave velocity is higher than that of the relaxed gastrocnemius muscle [6]. The lateral gastrocnemius muscle had a smaller shear wave velocity than the medial gastrocnemius muscle, but this is unlikely to be due to a difference in muscle length, and is assumed to be due to the pennation angle of muscle fibers or muscle fiber length. The similar difference in shear modulus was observed in healthy volunteers between the lateral and medial gastrocnemius muscle [6].

Increase of muscle stiffness in active muscle conditions is reported to have several causes, i.e. reflex-mediated stiffness by prolonged muscle contraction after repetitive task which lead

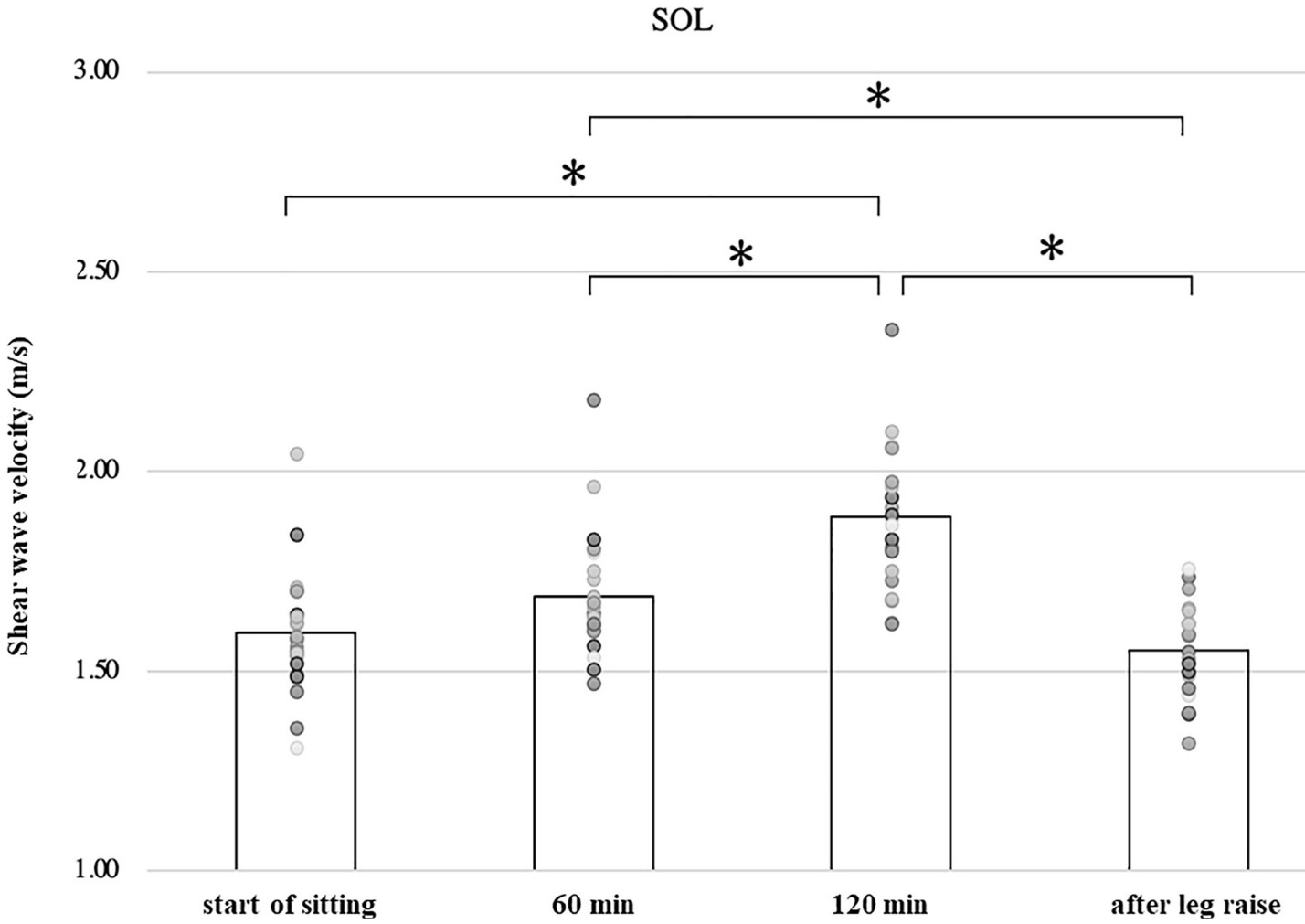

**Fig 7. Shear wave velocity of the soleus.**

to the release of metabolites and activate nociceptors [26, 27], temporary increase of muscle stiffness after repetitive muscle contraction due to static or dynamic exercises, which is reflected by increased intramuscular pressure associated with increases in intramuscular blood flow [28, 29], or static stretch of the muscles by which muscle stiffness increase proportionally to the muscle length [6]. Increase of muscle stiffness in resting muscle conditions is reported to have causes, i.e. increasing collagen mass in muscles by aging and spastic conditions with neuromuscular diseases [7, 30].

On the other hand, although no measurement by shear wave elastography has been performed, the another suspected causes of increase in muscle stiffness in resting muscles are reported to be fluid retention in the leg due to occupational leg edema after prolonged standing and sitting [12], long-haul flight simulation of sitting 4 to 12 hours [16], and the effect of 4 ours sitting and calf activity [17]. Belczak et al (2018) reported significant increase in leg volume by volumetric measurement using water displacement technique [12]. Mittermayr et al (2007) reported increased lower leg volume by plethysmographic measurement using an optoelectronic scanner system, suggesting the phenomena was due to extravascular fluid shifts in the lower leg. They also measured by ultrasonography the cross-sectional diameter of calf veins, there was no significant change in the diameter by 4 to 12 hours sitting [16]. Singh et al

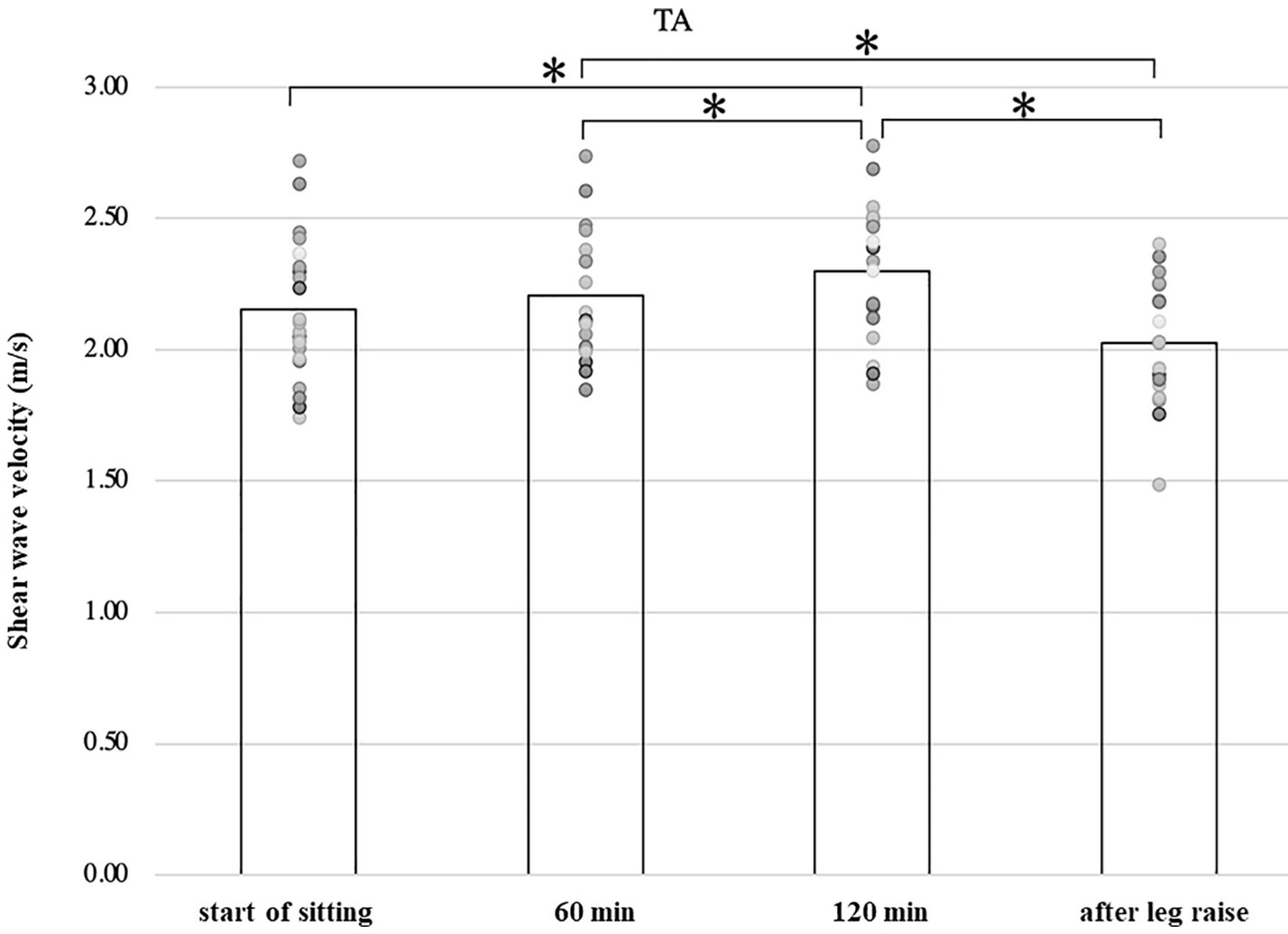

**Fig 8. Shear wave velocity of the tibialis anterior muscle.**

(2017) reported increased leg volume estimated by calf circumference and by bioelectrical impedance analysis. In their study, they suggested two points, first increase in immediate intra-vascular leg fluid due to hydrostatic pressure and a loss of leg muscle tone, second retention of extravascular fluid in the legs by hydrostatic pressure after 4 hours sitting [17].

Based on literature and result of our study, increase in shear wave velocity with prolonged sitting does not come from change of leg muscle fibers or connective tissue but may come from increase in intra-compartment pressure of the lower leg due to fluid retention in extra-cellular space.

It is presumed that the shear wave velocity decreased by leg elevation without delay because of the drainage of intra-venous blood out of lower leg, resulting in a decrease in intramuscular pressure (Table 1) [11]. This means that the blood in the muscle compartment is drained out. Even if the venous valves work in the lower leg in healthy adults, gradual leaking of the blood exudate might occur, resulting in increase in intramuscular pressure with time. However, at this point, we have not measured intramuscular pressure or venous blood flow, therefore it is only possible to estimate the change in internal pressure in the gastrocnemius and tibialis ante-rior muscle by changes in shear wave velocity.

During a long sitting period in a recliner, the knee joint is often kept at nearly 90 degrees of flexion and the ankle joint is kept in slight flexion. The results of this study have clinical importance in the possibility of prevention of muscle stiffness and edema of the lower legs during long time flight or evacuation in the car as refugees after huge earth quake. Measurement of shear wave velocity by non-invasive ultrasonography help people know the extent of leg edema in timely fashion [14, 31]. In such cases, it is important to know that simple elevation of lower leg for 3 minutes may reduce leg edema.

## Research limitations

**Measurement of muscle at rest.** In our study, we measured the leg immobile in the sitting position for a long time. However, except in special cases, it is rare to have a situation in which the subjects do not move his or her legs for 2 hours during sitting. In the future, it is necessary to make ultrasound measurements that take the motion of the lower limbs into account.

**Measurement for adult male.** In this study, only adult males were included in the measurements, but venous blood retention originally occurs more frequently in middle aged women and is observed as varicose veins and leg edema [11]. In the future, with the approval of the ethics committee, we will add measurements on the lower legs of aged women.

**Measurement up to 2 hours.** In this study, we adopted the lower leg sitting condition for up to 2 hours. According to previous reports, the occurrence of economy class syndrome is known to occur after a flight longer than 6 hours. While avoiding the development of venous thrombosis, measurements should also be performed under conditions exceeding 2 hours [14, 31].

No intramuscular pressure measurement is performed.

In this measurement, the internal pressure of the leg muscle compartment was estimated as the change in the value of the shear wave velocity. For more direct measurement, intramuscular pressure measurement is required [4, 25].

**Venous hemodynamics.** In the present measurement, blood retention in the vein was not assessed. Since ultrasonic measuring devices can measure the increase in the diameter of the intramuscular veins and blood flow, there is a need to investigate the hemodynamics of venous blood in conjunction with the measurement of shear wave velocity [11, 24].

## Conclusions

Shear wave velocity of the lower leg muscles increased with time in 2 hours sitting, and decreased with subsequent leg elevation. Based on the report that the change in the shear wave velocity was proportional to the internal pressure of the leg muscle compartment in turkey models, it is estimated that increase in shear wave velocity with prolonged sitting may come from increase in intra-compartment pressure of the lower leg due to fluid retention in extracellular space of the compartment.

## Supporting information

**S1 Table. Shear wave velocity of the lower leg muscles (n = 24).**
(DOCX)

## Acknowledgments

We express special thanks to all contributors (Tomoya Hayashi, Hisashi Honma, Yuji Sasaki, Kazuyuki Sugawara) for assistance of experiments.

## Author Contributions

**Conceptualization:** Kumiko Okino, Mitsuhiro Aoki.

**Data curation:** Kumiko Okino, Masahiro Yamane.

**Formal analysis:** Masahiro Yamane.

**Funding acquisition:** Mitsuhiro Aoki, Chikashi Kohmura.

**Investigation:** Kumiko Okino, Mitsuhiro Aoki, Chikashi Kohmura.

**Methodology:** Kumiko Okino, Mitsuhiro Aoki, Masahiro Yamane.

**Supervision:** Mitsuhiro Aoki, Chikashi Kohmura.

**Validation:** Mitsuhiro Aoki.

**Visualization:** Kumiko Okino.

**Writing – original draft:** Kumiko Okino.

**Writing – review & editing:** Mitsuhiro Aoki.

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
