## [Decision Letter · Decision Letter 0]

9 Feb 2021

PONE-D-20-36699

Effect of prolonged sitting immobility on shear wave velocity of the lower leg muscles

PLOS ONE

Dear Dr. Aoki,

Thank you for submitting your manuscript to PLOS ONE. After careful consideration, we feel that it has merit but does not fully meet PLOS ONE’s publication criteria as it currently stands. Therefore, we invite you to submit a revised version of the manuscript that addresses the points raised during the review process.

Although mainly descriptive your results might be of value to the community.

We look forward to receiving your revised manuscript.

Kind regards,

Guy Cloutier, Ph.D.

Academic Editor

PLOS ONE

Journal Requirements:

3.We note that Figure 2a includes an image of a patient / participant in the study. 

Reviewers' comments:

Reviewer's Responses to Questions

**Comments to the Author**

1. Is the manuscript technically sound, and do the data support the conclusions?

Reviewer #1: Yes

Reviewer #2: No

Reviewer #3: Partly

2. Has the statistical analysis been performed appropriately and rigorously? 

Reviewer #1: Yes

Reviewer #2: Yes

Reviewer #3: Yes

3. Have the authors made all data underlying the findings in their manuscript fully available?

Reviewer #1: Yes

Reviewer #2: Yes

Reviewer #3: No

4. Is the manuscript presented in an intelligible fashion and written in standard English?

Reviewer #1: No

Reviewer #2: Yes

Reviewer #3: Yes

5. Review Comments to the Author

Reviewer #1: General comments :

This study assessed shear wave velocity of lower limb muscles (gastrocnemius, soleus and tibialis anterior muscles) at baseline, after prolonged sitting (60 min and 120 min) and in a leg-raised position. The study seems well-conducted with a sufficient sample size and an adequate methodology. However, the manuscript would benefit being reviewed by a professional proofreader/editor to improve the language.

Specific comments:

P4-5: Please describe the inclusion and exclusion criteria. The information about where and how the participants were recruited should be provided. The characteristics of the participants (Table 1) should be moved to the result section. The related statistics and the test used to assess the normality of the data should be detailed in the statistical section of the methodology.

P5, L122: the number of participants should be in the participant section of the methodology or the results section.

P5, L128: the reference to the Figure 1 is not related to the text “The temperature and humidity in the laboratory were maintained at 25°C and 30-40%, respectively”.

P7, L167-168: Positioning of the probe on the gastrocnemius, soleus and tibialis anterior muscles. Which landmarks were used to establish the positioning of the probe (e.g. 30% and 50% of the leg). How was the muscle belly located?

P7: Did the participants remain in the same position for the elastography measurements? It is not clear that the surface scanned was easily accessible for the evaluator according the image provided.

P8: Surface electromyography. Can you further describe how the EMG data were managed during the entire testing period? Were the EMG traces visible for both the participants and the evaluator? Was a pre-established resting activity targeted on a screen for ensuring proper relaxation?

P9, L235: Replace “the risk rate was set at” by “level of significance was set at”

P 11 L283- 297: In the discussion, the values obtained should not be repeated. The results should rather be summarized and discussed according the current literature.

P12, L33-335: Do you have any reference supporting this hypothesis? The authors should explain the hypothesized mechanisms responsible for the increase in shear wave velocity can with prolonged sitting and they should support their explanations with the literature.

Reviewer #2: The reviewer has thoroughly reviewed the manuscript for scientific content, manuscript construction, and general impact. This study seems a sound one but needs to be redefined, especially in terms of novelty, hypothesis, and methodology. The main issues are listed below.

Introduction

The introduction section should be clear and hypothesis-driven to allow the readers to see the basis of the authors’ hypothesis. Although the hypothesis is stated in the manuscript, the reviewer cannot understand the rationale for the hypothesis. Additionally, in the introduction section, the authors should not exhaustively review the subject but need to provide relevant references. Namely, the authors should mention the literature on muscle conditions during a prolonged sitting.

The authors consider that the clinical importance of the present study is related to compartment syndrome, it would be better to measure intramuscular pressure instead of shear wave velocity.

Methods

The authors report 3ρVs2 because they regarded the muscles as isotropic (Line 190-). However, this is wrong (although the authors are aware).

The orientation of the ultrasound probe is unclear. As far as the reviewer see Figure 4, it seems that the probe was not aligned with the fascicle direction of the muscles. As the authors know, the probe orientation relative to fascicle direction affects the shear wave speed measured, leading to smaller shear wave velocity. In fact, the shear wave velocity reported in this paper is smaller than those in previous studies.

The ICC in this study is smaller than that of previous studies (ICC > 0.9).

The authors stated that no muscle contraction was observed to occur through EMG recording. Please explain in more detail what value the authors determined to be no muscle contractions.

Discussion

The discussion section needs to reflect what the authors found, how it relates to the literature, and then what it means clinically or physiologically. Each paragraph should be logical in sequence as at present; it is a bit hard to follow. Further, the discussion is qualitative but not quantitative.

Conclusion

Intramuscular pressure was not directly measured in this paper. Thus, the current conclusion is not based on the data of the present study.

Reviewer #3: PONE-D-20-36699

This study brings descriptive data to the literature on the effect of prolonged sitting on lower limb muscle shear wave velocity (i.e., muscle stiffness under known hypotheses of localized homogeneous, isotropic and purely linear elastic property, which oversimplify the muscle rheology but is of common use in the field). If the angle of insonification with respect to the muscle long axis is defined, this can bring informative data to the literature. The study protocol is quite simple and the sample size is limited in term of number and population characteristics; only normal young male volunteers were scanned. EMG verification of the absence of muscle contraction, which is an important confounder, was made to improve robustness of reported results. Overall, even though it is mainly a descriptive study with no clear rationale or hypothesis, reported results might be of value to design future studies on pathological cohorts of patients with musculoskeletal disorders. A strong effort in restructuring the paper is required, many sections are not at the right place into the manuscript and repetitions are noticed. Please use standard scientific report guidelines.

1) By considering the limited sample size, I would add “a proof-of-concept study” at the end of the title.

2) Abstract, L38-39: There is repetitive information on statistics.

3) Introduction, L76: The concept of anisotropic modulus is confusing. L73 is giving the Young’s modulus estimate for a linear, isotropic and purely elastic material, whereas the equation of L76 is simply the shear modulus under same hypotheses. There is no anisotropic modulus leading to this simplified equation, as far as I know.

4) L87: The concept of compartmental pressure increase is indeed of interest to interpret the dataset as the boundary condition imposed by the internal pressure on muscles changes the non-linear strain-stress relation of most biological tissues. Any pre-compressed tissue will result in a displacement of the measured elasticity on the non-linear strain-stress curve and thus will lead to a different estimate of the Young’s modulus given by the equation on L73. This is a limitation of shear wave elastography imaging as linear elasticity is assumed. It would be important to bring basic tissue mechanics concepts into your discussion to avoid misleading interpretation.

5) L97: Your hypothesis is empty and not based on strong rationales. This should be improved. Why prolonged sitting would change muscle stiffness?

6) Table 1 is useless and should be removed.

7) L159: “precision”.

8) L186 and 188: What is a “shear wave verbosity” and what is “a constant condition”?

9) L203: What is the output of the Web-1000 system?

10) Discussion: Remove all repetitions on reported results, this is useless.

11) L309 and elsewhere: Correct “verbosity”.

12) L315: Are you sure about this curious hypothesis? How the hydrostatic pressure applied on muscles can be induced by the venous low pressure system of typically a few mmHg? This sounds curious to me.

13) L317: You are overstretching the interpretation of your results since you did not measure the intramuscular pressure; consequently you cannot say that this is “in accordance with”.

14) L327: The muscle length has certainly nothing to do with shear wave speed; you are likely referring to the muscle stretching.

15) L328: “nature of muscle”, can you be more descriptive and support your hypothesis based on known histopathology analysis?

16) L330: Here again, verify your hypothesis about venous pressure and impact on muscle. Swelling of muscle versus direct impact of venous pressure are likely different mechanisms. L332: Even though you are citing the literature, more rationale should be given on how an intravenous pressure of typical < 10 mmHg would compress the muscle enough to change the strain-stress non-linear elasticity behavior.

17) L342: Indeed assessing the venous system with Doppler flow and B-mode diameter measurements would have helped supporting your hypothesis; otherwise your study is mainly descriptive. An impedance plethysmography system would also help assessing electrolyte volume (i.e., blood volume and interstitial inter-compartment water).

18) Conclusion: Here again you are overstretching the interpretation of your results; no internal pressure has been reported.

19) L391: “evacuation in the car”, what do you mean?

6. PLOS authors have the option to publish the peer review history of their article (what does this mean?). If published, this will include your full peer review and any attached files.

Reviewer #1: No

Reviewer #2: No

Reviewer #3: No

---

## [Author Response · Author response to Decision Letter 0]

17 Mar 2021

Reply to the Journal requirement: 

The title of the manuscript was revised in all uploaded documents, including cover letter, manuscript, and rebuttal letter as follows: “Effect of prolonged sitting immobility on shear wave velocity of the lower leg muscles in healthy adults: a proof-of-concept study”

Line117, (Line119-124): All ethics statements of participants were moved to the Method section.

Line119, (Line121-122): Statements of consent for publication of the individual who appeared in the text was provided as follows: “The individual in this manuscript has given written informed consent to publish these case details.”

Following marks, Line**-**, (Line**- **), suggests Line on the original manuscript, and (Line on the revised manuscript).

Reply to Reviewer #1, #2, #3

Reviewer #1: General comments:

This study assessed shear wave velocity of lower limb muscles (gastrocnemius, soleus and tibialis anterior muscles) at baseline, after prolonged sitting (60 min and 120 min) and in a leg-raised position. The study seems well-conducted with a sufficient sample size and an adequate methodology. However, the manuscript would benefit being reviewed by a professional proofreader/editor to improve the language.

Specific comments:

P4-5: Please describe the inclusion and exclusion criteria. The information about where and how the participants were recruited should be provided. The characteristics of the participants (Table 1) should be moved to the result section. The related statistics and the test used to assess the normality of the data should be detailed in the statistical section of the methodology.

Line105, (Line112-113): Information about recruitment of participant was stated in the Method section as follows: “Subjects were recruited by announcements on posters which were exhibited in a student bulletin board.” 

Table 1 was removed. 

Line110, (Line236-237): Normality of the data was moved to statistics section.

P5, L122: the number of participants should be in the participant section of the methodology or the results section.

Line122: Number of participants was removed.

P5, L128: the reference to the Figure 1 is not related to the text “The temperature and humidity in the laboratory were maintained at 25°C and 30-40%, respectively”.

Line128, (Line131): Reference of (Figure.1) was moved up to adequate location.

P7, L167-168: Positioning of the probe on the gastrocnemius, soleus and tibialis anterior muscles. Which landmarks were used to establish the positioning of the probe (e.g. 30% and 50% of the leg). How was the muscle belly located?

Line 167-171, (Line169-178): The prove positioning and location on the muscle belly was stated specifically as follows: “The location of a probe to be applied on the muscle belly of the medial and lateral gastrocnemius, soleus and tibialis anterior muscles was adjusted in the designated position (the medial and lateral portions of the gastrocnemius muscle at 30 % of the proximal left lower leg, the soleus muscle at 50 % of the left lower leg, and the tibialis anterior muscle at 30 % of the proximal left lower leg [2, 5]) for individual subject. On the most prominent part of the muscle belly, the rectangular shape skin mark with adhesive tape was attached prior to the measurement, and the probe was placed in the rectangular mark (Fig 3a, 3b). Since orientation of muscle fascicles of the gastrocnemius, soleus is oblique by their pennated structure and is redundant due to muscle relaxation, prove setting was aligned to anatomical muscle shortening direction.” 

P7: Did the participants remain in the same position for the elastography measurements? It is not clear that the surface scanned was easily accessible for the evaluator according the image provided.

Line 142, (Line154-155): “Participants remained in the same position sitting in a chair from the beginning until the end of the elastography measurement, except 3 minutes leg elevation.” This statement was added at the end of “Elevation of the lower limb”.

Line168-171, (Line169-178): A prove positioning and location of the muscle belly was stated specifically as follows: “The location of a probe to be applied on the muscle belly of the medial and lateral gastrocnemius, soleus and tibialis anterior muscles was adjusted in the designated position (the medial and lateral portions of the gastrocnemius muscle at 30 % of the proximal left lower leg, the soleus muscle at 50 % of the left lower leg, and the tibialis anterior muscle at 30 % of the proximal left lower leg [2, 5]) for individual subject. On the most prominent part of the muscle belly, the rectangular shape skin mark with adhesive tape was attached prior to the measurement, and the probe was placed in the rectangular mark (Fig 3a, 3b). Since orientation of muscle fascicles of the gastrocnemius, soleus is oblique by their pennated structure and is redundant due to muscle relaxation, prove setting was aligned to anatomical muscle shortening direction.”

.

P8: Surface electromyography. Can you further describe how the EMG data were managed during the entire testing period? Were the EMG traces visible for both the participants and the evaluator? Was a pre-established resting activity targeted on a screen for ensuring proper relaxation?

Line203, (Line208-212): Statement of traceability of EMG data during elastography measurement for the evaluator was added as follows: “During shear wave velocity measurement by ultra-sonography, EMG data was continuously displayed for examiner and subject on PC screen, suggesting the meaning of appearance of muscle activation waves for ensuring proper relaxation.”

P9, L235: Replace “the risk rate was set at” by “level of significance was set at”

Line235, (Line245): “the risk rate was set at” was replaced by “level of significance was set at” as suggested.

P 11 L283- 297: In the discussion, the values obtained should not be repeated. The results should rather be summarized and discussed according the current literature.

L283- 297, (Line287-373): All repetitions on reported results were removed from the Discussion

P12, L333-335: Do you have any reference supporting this hypothesis? The authors should explain the hypothesized mechanisms responsible for the increase in shear wave velocity can with prolonged sitting and they should support their explanations with the literature.

Line97, (Line102-106): The hypothesis was added by providing specific pathology in the lower leg muscles.

Line333-335, (Line331-358): The hypothesized mechanism was provided in two points of view, i.e. one is change of leg muscle fibers and the other is change of extra-cellular space in the compartment. 

Reviewer #2: The reviewer has thoroughly reviewed the manuscript for scientific content, manuscript construction, and general impact. This study seems a sound one but needs to be redefined, especially in terms of novelty, hypothesis, and methodology. The main issues are listed below.

Introduction

The introduction section should be clear and hypothesis-driven to allow the readers to see the basis of the authors’ hypothesis. Although the hypothesis is stated in the manuscript, the reviewer cannot understand the rationale for the hypothesis. Additionally, in the introduction section, the authors should not exhaustively review the subject but need to provide relevant references. Namely, the authors should mention the literature on muscle conditions during a prolonged sitting.

The authors consider that the clinical importance of the present study is related to compartment syndrome, it would be better to measure intramuscular pressure instead of shear wave velocity.

Line97, (Line102-106): The hypothesis was added by providing specific pathology in the lower leg muscles as follows: “We hypothesized that the shear wave velocity of the medial and lateral gastrocnemius, soleus and tibialis anterior muscle would increase with time in the resting leg position after 2 hours of sitting and decrease with subsequent leg elevation. Then we considered the mechanism of the velocity change with two points of view, i.e. one is change of leg muscle fibers and the other is change of extra-cellular space in the compartment.”

Line91, (Line91-96): In the middle of Introduction, estimated muscle condition during a prolonged sitting was listed by mentioning literature as follows: “There are several clinical conditions that cause lower leg symptom during a prolonged sitting, i.e. occupational leg edema after prolonged standing and sitting [12], thrombophlebitis of lower leg during long flight [13, 14] and fluid accumulation in the leg during long-distance bus travel [15]. To solve mechanism of prolonged sitting in lower leg, plethysmographic measurement for haemodynamics of calf veins or bioelectrical impedance analysis for tissue fluid has been performed [16, 17], but no measurement of resting leg muscles by shear wave elastography has been reported.”

”. 

Line329-, (Line331-339): In Discussion, to verify hypothesis about venous pressure and impact on muscle, speculated pathology occurred in the lower leg muscles was stated with providing literature as follows: “Increase of muscle stiffness in active muscle conditions is reported to have several causes, i.e. reflex-mediated stiffness by prolonged muscle contraction after repetitive task which lead to the release of metabolites and activate nociceptors [26, 27], temporary increase of muscle stiffness after repetitive muscle contraction due to static or dynamic exercises, which is reflected by increased intramuscular pressure associated with increases in intramuscular blood flow [28, 29], or static stretch of the muscles by which muscle stiffness increase proportionally to the muscle length [6]. Increase of muscle stiffness in resting muscle conditions is reported to have causes, i.e. increasing collagen mass in muscles by aging and spastic conditions with neuromuscular diseases [7, 30].” 

“On the other hand, although no measurement by shear wave elastography has been performed, the another suspected causes of increase in muscle stiffness in resting muscles are reported to be fluid retention in the leg due to occupational leg edema after prolonged standing and sitting [12], long-haul flight simulation of sitting 4 to 12 hours [16], and the effect of 4 ours sitting and calf activity [17]. Belczak et al (2018) reported significant increase in leg volume by volumetric measurement using water displacement technique [12]. Mittermayr et al (2007) reported increased lower leg volume by plethysmographic measurement using an optoelectronic scanner system, suggesting the phenomena was due to extravascular fluid shifts in the lower leg. They also measured by ultrasonography the cross-sectional diameter of calf veins, there was no significant change in the diameter by 4 to 12 hours sitting [16]. Singh et al (2017) reported increased leg volume estimated by calf circumference and by bioelectrical impedance analysis. In their study, they suggested two points, first increase in immediate intravascular leg fluid due to hydrostatic pressure and a loss of leg muscle tone, second retention of extravascular fluid in the legs by hydrostatic pressure after 4 hours sitting [17].” 

Line344, (Line356-358): The following summarized statements was inserted. 

“Based on literature and result of our study, increase in shear wave velocity with prolonged sitting does not come from change of leg muscle fibers or connective tissue but may come from increase in intra-compartment pressure of the lower leg due to fluid retention in extra-cellular space.” 

Line347, (Line370-373): Clinical importance was stated about detection and treatment of leg edema with measurement of shear wave velocity by non-invasive ultrasonography as follows: “The results of this study have clinical importance in the possibility of prevention of muscle stiffness and edema of the lower legs during long time flight or evacuation in the car as refugees after huge earth quake. Measurement of shear wave velocity by non-invasive ultrasonography help people know the extent of leg edema in timely fashion.” 

Methods

The authors report 3ρVs2 because they regarded the muscles as isotropic (Line 190-). However, this is wrong (although the authors are aware).

The orientation of the ultrasound probe is unclear. As far as the reviewer see Figure 4, it seems that the probe was not aligned with the fascicle direction of the muscles. As the authors know, the probe orientation relative to fascicle direction affects the shear wave speed measured, leading to smaller shear wave velocity. In fact, the shear wave velocity reported in this paper is smaller than those in previous studies.

The ICC in this study is smaller than that of previous studies (ICC > 0.9).

The authors stated that no muscle contraction was observed to occur through EMG recording. Please explain in more detail what value the authors determined to be no muscle contractions.

Line167-171, (Line169-178): Detail placement of probe and location on the muscle belly was stated in the text as follows: “The location of a probe to be applied on the muscle belly of the medial and lateral gastrocnemius, soleus and tibialis anterior muscles was adjusted in the designated position (the medial and lateral portions of the gastrocnemius muscle at 30 % of the proximal left lower leg, the soleus muscle at 50 % of the left lower leg, and the tibialis anterior muscle at 30 % of the proximal left lower leg [2, 5]) for individual subject. On the most prominent part of the muscle belly, the rectangular shape skin mark with adhesive tape was attached prior to the measurement, and the probe was placed in the rectangular mark (Fig 3a, 3b). Since orientation of muscle fascicles of the gastrocnemius, soleus is oblique by their pennated structure and is redundant due to muscle relaxation, prove setting was aligned to anatomical muscle shortening direction.” 

Line203-204, (Line211-212): Statement of EMG activity recording was corrected as follows: “Appearance of EMG activity was determined by observation of action potentials greater than 2SD amplitude of baseline.”

Discussion

The discussion section needs to reflect what the authors found, how it relates to the literature, and then what it means clinically or physiologically. Each paragraph should be logical in sequence as at present; it is a bit hard to follow. Further, the discussion is qualitative but not quantitative.

Line329-, (Line331-358): We added in Discussion hypothesized mechanism with two points of view, i.e. one is change of leg muscle fibers and the other is change of extra-cellular space in the compartment. 

Conclusion

Intramuscular pressure was not directly measured in this paper. Thus, the current conclusion is not based on the data of the present study.

Line384-391, (Line403-407): Conclusion was revised based on the result of increase in shear wave velocity and speculated pathology in the lower leg muscle as follows: “Shear wave velocity of the lower leg muscles increased with time in 2 hours sitting, and decreased with subsequent leg elevation. It has been reported that the change in the shear wave velocity was proportional to the internal pressure of the leg muscle compartment. Increase in shear wave velocity with prolonged sitting may come from increase in intra-compartment pressure of the lower leg due to fluid retention in extra-cellular space of the compartment.” 

Line43-44, (Line45): Conclusion in Abstract was revised based on Conclusion in the Text as follows: “and it is assumed that the increase of shear wave velocity in the 2-hour seated leg is due to fluid retention in extra-cellular space of the compartment.”

Reviewer #3: PONE-D-20-36699

This study brings descriptive data to the literature on the effect of prolonged sitting on lower limb muscle shear wave velocity (i.e., muscle stiffness under known hypotheses of localized homogeneous, isotropic and purely linear elastic property, which oversimplify the muscle rheology but is of common use in the field). If the angle of insonification with respect to the muscle long axis is defined, this can bring informative data to the literature. The study protocol is quite simple and the sample size is limited in term of number and population characteristics; only normal young male volunteers were scanned. EMG verification of the absence of muscle contraction, which is an important confounder, was made to improve robustness of reported results. Overall, even though it is mainly a descriptive study with no clear rationale or hypothesis, reported results might be of value to design future studies on pathological cohorts of patients with musculoskeletal disorders. A strong effort in restructuring the paper is required, many sections are not at the right place into the manuscript and repetitions are noticed. Please use standard scientific report guidelines.

1) By considering the limited sample size, I would add “a proof-of-concept study” at the end of the title.

Line1, (Line1): “a proof-of-concept study” was added in the title as suggested.

2) Abstract, L38-39: There is repetitive information on statistics.

Line30-39, (Line39-40): The repetitive part was removed and data in parenthesis was corrected.

3) Introduction, L76: The concept of anisotropic modulus is confusing. L73 is giving the Young’s modulus estimate for a linear, isotropic and purely elastic material, whereas the equation of L76 is simply the shear modulus under same hypotheses. There is no anisotropic modulus leading to this simplified equation, as far as I know.

4) L87: The concept of compartmental pressure increase is indeed of interest to interpret the dataset as the boundary condition imposed by the internal pressure on muscles changes the non-linear strain-stress relation of most biological tissues. Any pre-compressed tissue will result in a displacement of the measured elasticity on the non-linear strain-stress curve and thus will lead to a different estimate of the Young’s modulus given by the equation on L73. This is a limitation of shear wave elastography imaging as linear elasticity is assumed. It would be important to bring basic tissue mechanics concepts into your discussion to avoid misleading interpretation.

Line75-76, (Line75-79): The sentence “On the contrary, if the density of muscles is not uniform, it is expressed in terms of the anisotropic modulus (anisotropic E= ρVs2) [2,7].” was removed to avoid confusion. And the paragraph was revised as follows, “Regarding the difference between elastic modulus and shear wave velocity, Eby (2013) found that the equation E=3ρVs2 (E: elastic modulus kPa, Vs: shear wave velocity m/s, ρ: muscle density 1000 kg/m3) holds if the density of the muscle is assumed to be uniform, in case of measurement along longitudinal muscle fiber direction [4]. On the contrary, if the density of muscles is not uniform, in case of measurement along transverse or oblique muscle fiber direction, it is expressed in terms of the anisotropic modulus (E=ρVs2) [2,7]. In researches of the shear wave velocity in the leg muscles at a resting state (ankle position around the slack angle without muscle contraction), the muscle tissue was measured regarding as the isotrophic modulus (E= 3ρVs2) [2,5,7-9].” 

5) L97: Your hypothesis is empty and not based on strong rationales. This should be improved. Why prolonged sitting would change muscle stiffness?

Line97, (Line102-106): The hypothesis was added by providing specific pathology in the lower leg muscles as follows: “We hypothesized that the shear wave velocity of the medial and lateral gastrocnemius, soleus and tibialis anterior muscle would increase with time in the resting leg position after 2 hours of sitting and decrease with subsequent leg elevation. Then we considered the mechanism of the velocity change with two points of view, i.e. one is change of leg muscle fibers and the other is change of extra-cellular space in the compartment.”

6) Table 1 is useless and should be removed.

Table 1 was removed and detail statement of this result was provided in the Results.

7) L159: “precision”.

Line159: typo error was corrected

8) L186 and 188: What is a “shear wave verbosity” and what is “a constant condition”?

Line186: “a constant” was removed as suggested. 

9) L203: What is the output of the Web-1000 system?

Line203, (Line206-208): The sentence was corrected as follows: “The electromyography data was recorded at 2000 Hz and collected to a polygraph system of Web-1000 (Nihon Koden Ltd., Tokyo, Japan), then analyzed using LabChart version 7 (AD Instruments, Tokyo, Japan).”

10) Discussion: Remove all repetitions on reported results, this is useless.

L283- 297, (Line287-373): All repetitions on reported results were removed from the Discussion

11) L309 and elsewhere: Correct “verbosity”.

Line309: All typo error was corrected

12) L315: Are you sure about this curious hypothesis? How the hydrostatic pressure applied on muscles can be induced by the venous low pressure system of typically a few mmHg? This sounds curious to me.

Line309-320, (Line314-319): These two paragraphs were revised to state the effect of hydrostatic pressure in the low leg with sitting position to avoid confusions as follows: “In sitting position, venous pressure of popliteal vein at the popliteal fossa and pressure of saphenous vein at the medial malleolus were reported to be 23.9 mmHg (3.17 kPa) and 54.2 mmHg (7.2 kPa), respectively [25]. Considering the location of shear wave velocity measurements in the tibialis anterior and gastrocnemius muscles in the sitting position, it can be inferred that hydrostatic pressure is equally applied to the tibialis anterior and gastrocnemius muscles through the leg veins for two hours.”

13) L317: You are overstretching the interpretation of your results since you did not measure the intramuscular pressure; consequently you cannot say that this is “in accordance with”.

Line317: This paragraph was removed.

14) L327: The muscle length has certainly nothing to do with shear wave speed; you are likely referring to the muscle stretching.

Line327, (Line326): We consider the statement of muscle length in this context was correct. 

15) L328: “nature of muscle”, can you be more descriptive and support your hypothesis based on known histopathology analysis?

Line328, (Line327): The phrase “nature of muscle” was replaced by “pennation angle of muscle fibers or muscle fiber length” 

16) L330: Here again, verify your hypothesis about venous pressure and impact on muscle. Swelling of muscle versus direct impact of venous pressure are likely different mechanisms. L332: Even though you are citing the literature, more rationale should be given on how an intravenous pressure of typical < 10 mmHg would compress the muscle enough to change the strain-stress non-linear elasticity behavior.

Line329-, (Line331-358): To verify hypothesis about venous pressure and impact on muscle, speculated pathology occurred in the lower leg muscles was stated with providing literature as follows: “Increase of muscle stiffness in active muscle conditions is reported to have several causes, i.e. reflex-mediated stiffness by prolonged muscle contraction after repetitive task which lead to the release of metabolites and activate nociceptors [26, 27], temporary increase of muscle stiffness after repetitive muscle contraction due to static or dynamic exercises, which is reflected by increased intramuscular pressure associated with increases in intramuscular blood flow [28, 29], or static stretch of the muscles by which muscle stiffness increase proportionally to the muscle length [6]. Increase of muscle stiffness in resting muscle conditions is reported to have causes, i.e. increasing collagen mass in muscles by aging and spastic conditions with neuromuscular diseases [7, 30].” 

“On the other hand, although no measurement by shear wave elastography has been performed, the another suspected causes of increase in muscle stiffness in resting muscles are reported to be fluid retention in the leg due to occupational leg edema after prolonged standing and sitting [12], long-haul flight simulation of sitting 4 to 12 hours [16], and the effect of 4 ours sitting and calf activity [17]. Belczak et al (2018) reported significant increase in leg volume by volumetric measurement using water displacement technique [12]. Mittermayr et al (2007) reported increased lower leg volume by plethysmographic measurement using an optoelectronic scanner system, suggesting the phenomena was due to extravascular fluid shifts in the lower leg. They also measured by ultrasonography the cross-sectional diameter of calf veins, there was no significant change in the diameter by 4 to 12 hours sitting [16]. Singh et al (2017) reported increased leg volume estimated by calf circumference and by bioelectrical impedance analysis. In their study, they suggested two points, first increase in immediate intravascular leg fluid due to hydrostatic pressure and a loss of leg muscle tone, second retention of extravascular fluid in the legs by hydrostatic pressure after 4 hours sitting [17]” 

Line344, (Line356-358): The following summarized statements was inserted. 

“Based on literature and result of our study, increase in shear wave velocity with prolonged sitting does not come from change of leg muscle fibers or connective tissue but may come from increase in intra-compartment pressure of the lower leg due to fluid retention in extra-cellular space.

” 

17) L342: Indeed assessing the venous system with Doppler flow and B-mode diameter measurements would have helped supporting your hypothesis; otherwise your study is mainly descriptive. An impedance plethysmography system would also help assessing electrolyte volume (i.e., blood volume and interstitial inter-compartment water).

Line342, (Line341-354): Providing literature, statement of plethysmographic measurement and bioelectrical impedance analysis for leg edema was added in Discussion. 

18) Conclusion: Here again you are overstretching the interpretation of your results; no internal pressure has been reported.

Line384-391, (Line403-407): Conclusion was revised based on the result of increase in shear wave velocity and speculated pathology in the lower leg muscle as follows: “Shear wave velocity of the lower leg muscles increased with time in 2 hours sitting, and decreased with subsequent leg elevation. It has been reported that the change in the shear wave velocity was proportional to the internal pressure of the leg muscle compartment. Increase in shear wave velocity with prolonged sitting may come from increase in intra-compartment pressure of the lower leg due to fluid retention in extra-cellular space of the compartment.”

19) L391: “evacuation in the car”, what do you mean?

Line391, (Line370-373): We added statement of evacuation in the car as refugees after huge earth-quake as follows: “The results of this study have clinical importance in the possibility of prevention of muscle stiffness and edema of the lower legs during long time flight or evacuation in the car as refugees after huge earth quake. Measurement of shear wave velocity by non-invasive ultrasonography help people know the extent of leg edema in timely fashion [14, 33]” 

At the time of big earthquake named “The east Japan big earthquake in 2011”, many refugees stayed in their own cars for weeks because of lack in housings. Significant number of evacuated people suffered from deep vein thrombosis.

---

## [Decision Letter · Decision Letter 1]

9 Apr 2021

PONE-D-20-36699R1

Effect of prolonged sitting immobility on shear wave velocity of the lower leg muscles in healthy adults: a proof-of-concept study

PLOS ONE

Dear Dr. Aoki,

Thank you for submitting your manuscript to PLOS ONE. After careful consideration, we feel that it has merit but does not fully meet PLOS ONE’s publication criteria as it currently stands. Therefore, we invite you to submit a revised version of the manuscript that addresses the points raised during the review process.

ACADEMIC EDITOR: I also reviewed your responses and I agree with Reviewer #3, a new round of revision with clear responses to all comments and associated changes into the manuscript should be provided. Take all advises given by this reviewer for not further delaying the decision.

We look forward to receiving your revised manuscript.

Kind regards,

Guy Cloutier, Ph.D.

Academic Editor

PLOS ONE

Journal Requirements:

Reviewers' comments:

Reviewer's Responses to Questions

**Comments to the Author**

1. If the authors have adequately addressed your comments raised in a previous round of review and you feel that this manuscript is now acceptable for publication, you may indicate that here to bypass the “Comments to the Author” section, enter your conflict of interest statement in the “Confidential to Editor” section, and submit your "Accept" recommendation.

Reviewer #3: (No Response)

2. Is the manuscript technically sound, and do the data support the conclusions?

Reviewer #3: Partly

3. Has the statistical analysis been performed appropriately and rigorously? 

Reviewer #3: Yes

4. Have the authors made all data underlying the findings in their manuscript fully available?

Reviewer #3: No

5. Is the manuscript presented in an intelligible fashion and written in standard English?

Reviewer #3: No

6. Review Comments to the Author

Reviewer #3: Overall, your responses to reviewers were badly formulated and it was very difficult to identify if all comments were properly answered. Each reviewer comment should be followed by a clear section indicating your response (e.g., RESPONSE TO COMMENT #1, RESPONSE TO COMMENT #2, etc ...). Also a different color code should be used for each reviewer to allow facilitating identifying changes made in the text. I could notice again the overstretching of the conclusion: "It has been reported that the change in the shear wave velocity was proportional to the internal pressure of the leg muscle compartment", you did not measure any internal pressure! Some typos were also noticed into changes made in the revised document.

No decision can be made unless a new "responses to reviewers" document is made with clear color-coded changes.

7. PLOS authors have the option to publish the peer review history of their article (what does this mean?). If published, this will include your full peer review and any attached files.

Reviewer #3: No

---

## [Author Response · Author response to Decision Letter 1]

19 Apr 2021

PONE-D-20-36699R1

Reply to the comments of Academic Editor in the first revision

The title of the manuscript was revised in all uploaded documents, including cover letter, manuscript, and rebuttal letter as follows: “Effect of prolonged sitting immobility on shear wave velocity of the lower leg muscles in healthy adults: a proof-of-concept study”

Line117, (Line121-125): All ethics statements of participants were moved to the Method section.

Line119, (Line121-122): Statements of consent for publication of the individual who appeared in the text was provided as follows: “The individuals in this manuscript have given written informed consent to publish obtained data including images of the participants.”

Changes of Reference in 1st revision were listed below: 

Ref #1-11 remained unchanged in 1st revision.

Ref#12-17, and Ref#18-19 were moved to Ref#18-23 and Ref#24-25 in 1st revision. 

Ref#20-21 were removed in 1st revision.

Ref#22-24 were moved to Ref#13-14, and Ref#33 in 1st revision.

Ref#12 was added in 1st revision.

Ref#15-17 were added in 1st revision.

Ref#26-30 were added in 1st revision.

Reply to comments of Academic Editor in second revision

Ref#31-32 in 1st revision were removed, because these two literatures were unnecessary in second revision. 

Supporting File, that contains detail contents of Table 1, was uploaded as S Table. In this file, unit, number of specimens, mean, median, standard deviation, standard error, minimum, maximum, coefficient of variation for each muscle were presented. 

Following marks, Line**-**, (Line**- **), suggests Line in original manuscript, and (Line in first revision).

Reply to comments of Reviewer #1, #2, and #3 in first revision and second revision

Reviewer #1: in the first revision

General comments:

This study assessed shear wave velocity of lower limb muscles (gastrocnemius, soleus and tibialis anterior muscles) at baseline, after prolonged sitting (60 min and 120 min) and in a leg-raised position. The study seems well-conducted with a sufficient sample size and an adequate methodology. However, the manuscript would benefit being reviewed by a professional proofreader/editor to improve the language.

Specific comments:

P4-5: Please describe the inclusion and exclusion criteria. The information about where and how the participants were recruited should be provided. The characteristics of the participants (Table 1) should be moved to the result section. The related statistics and the test used to assess the normality of the data should be detailed in the statistical section of the methodology.

Response of comment #1): Line105, (Line112-113): Information about recruitment of participant was stated in the Method section as follows: “Subjects were recruited by announcements on posters which were exhibited in a student bulletin board.” 

Response of comment #2): Table 1 was removed. 

Response of comment #3): Line110, (Line237-238): Normality of the data was moved to statistics section.

Response of comment #4): P5, L122: Number of participants was removed.

Response of comment #5): P5, L128: the reference to the Figure 1 is not related to the text 

(Line 133), “The temperature and humidity in the laboratory were maintained at 25°C and 30-40%, respectively”.

Response of comment #6): Line128, (Line132): Reference of (Figure.1) was moved up to adequate location.

P7, L167-168: Positioning of the probe on the gastrocnemius, soleus and tibialis anterior muscles. Which landmarks were used to establish the positioning of the probe (e.g. 30% and 50% of the leg). How was the muscle belly located?

Response of comment #7): Line 167-171, (Line170-179): The prove positioning and location on the muscle belly was stated specifically as follows: “The location of a probe to be applied on the muscle belly of the medial and lateral gastrocnemius, soleus and tibialis anterior muscles was adjusted in the designated position (the medial and lateral portions of the gastrocnemius muscle at 30 % of the proximal left lower leg, the soleus muscle at 50 % of the left lower leg, and the tibialis anterior muscle at 30 % of the proximal left lower leg [2, 5]) for individual subject. On the most prominent part of the muscle belly, the rectangular shape skin mark with adhesive tape was attached prior to the measurement, and the probe was placed in the rectangular mark (Fig 3a, 3b). Since orientation of muscle fascicles of the gastrocnemius, soleus is oblique by their pennated structure and is redundant due to muscle relaxation, prove setting was aligned to anatomical muscle shortening direction.” 

P7: Did the participants remain in the same position for the elastography measurements? It is not clear that the surface scanned was easily accessible for the evaluator according the image provided.

Response of comment #8): Line 142, (Line155-156): “Participants remained in the same position sitting in a chair from the beginning until the end of the elastography measurement, except 3 minutes leg elevation.” This statement was added at the end of “Elevation of the lower limb”.

Response of comment #9): Line168-171, (Line170-179): A prove positioning and location of the muscle belly was stated specifically as follows: “The location of a probe to be applied on the muscle belly of the medial and lateral gastrocnemius, soleus and tibialis anterior muscles was adjusted in the designated position (the medial and lateral portions of the gastrocnemius muscle at 30 % of the proximal left lower leg, the soleus muscle at 50 % of the left lower leg, and the tibialis anterior muscle at 30 % of the proximal left lower leg [2, 5]) for individual subject. On the most prominent part of the muscle belly, the rectangular shape skin mark with adhesive tape was attached prior to the measurement, and the probe was placed in the rectangular mark (Fig 3a, 3b). Since orientation of muscle fascicles of the gastrocnemius, soleus is oblique by their pennated structure and is redundant due to muscle relaxation, prove setting was aligned to anatomical muscle shortening direction.”

P8: Surface electromyography. Can you further describe how the EMG data were managed during the entire testing period? Were the EMG traces visible for both the participants and the evaluator? Was a pre-established resting activity targeted on a screen for ensuring proper relaxation?

Response of comment #10): Line203, (Line209-212): Statement of traceability of EMG data during elastography measurement for the evaluator was added as follows: “During shear wave velocity measurement by ultra-sonography, EMG data was continuously displayed for examiner and subject on PC screen, suggesting the meaning of appearance of muscle activation waves for ensuring proper relaxation.”

P9, L235: Replace “the risk rate was set at” by “level of significance was set at”

Response of comment #11): Line235, (Line246): “the risk rate was set at” was replaced by “level of significance was set at” as suggested.

P 11 L283- 297: In the discussion, the values obtained should not be repeated. The results should rather be summarized and discussed according the current literature.

Response of comment #12): L283- 297, (Line287-376): All repetitions on reported results were removed from the Discussion

P12, L333-335: Do you have any reference supporting this hypothesis? The authors should explain the hypothesized mechanisms responsible for the increase in shear wave velocity can with prolonged sitting and they should support their explanations with the literature.

Response of comment #13): Line97, (Line102-106): The hypothesis was added by providing specific pathology in the lower leg muscles.

Response of comment #14): Line333-335, (Line332-359): The hypothesized mechanism was provided in two points of view, i.e. one is change of leg muscle fibers and the other is change of extra-cellular space in the compartment. 

Reviewer #2: in the first revision

The reviewer has thoroughly reviewed the manuscript for scientific content, manuscript construction, and general impact. This study seems a sound one but needs to be redefined, especially in terms of novelty, hypothesis, and methodology. The main issues are listed below.

Introduction

The introduction section should be clear and hypothesis-driven to allow the readers to see the basis of the authors’ hypothesis. Although the hypothesis is stated in the manuscript, the reviewer cannot understand the rationale for the hypothesis. Additionally, in the introduction section, the authors should not exhaustively review the subject but need to provide relevant references. Namely, the authors should mention the literature on muscle conditions during a prolonged sitting.

The authors consider that the clinical importance of the present study is related to compartment syndrome, it would be better to measure intramuscular pressure instead of shear wave velocity.

Response of comment #1): Line97, (Line102-106): The hypothesis was added by providing specific pathology in the lower leg muscles as follows: “We hypothesized that the shear wave velocity of the medial and lateral gastrocnemius, soleus and tibialis anterior muscle would increase with time in the resting leg position after 2 hours of sitting and decrease with subsequent leg elevation. Then we considered the mechanism of the velocity change with two points of view, i.e. one is change of leg muscle fibers and the other is change of extra-cellular space in the compartment.”

Response of comment #2): Line91, (Line91-96): In the middle of Introduction, estimated muscle condition during a prolonged sitting was listed by mentioning literature as follows: “There are several clinical conditions that cause lower leg symptom during a prolonged sitting, i.e. occupational leg edema after prolonged standing and sitting [12], thrombophlebitis of lower leg during long flight [13, 14] and fluid accumulation in the leg during long-distance bus travel [15]. To solve mechanism of prolonged sitting in lower leg, plethysmographic measurement for haemodynamics of calf veins or bioelectrical impedance analysis for tissue fluid has been performed [16, 17], but no measurement of resting leg muscles by shear wave elastography has been reported.”

Response of comment #3): Line329-, (Line332-359): In Discussion, to verify hypothesis about venous pressure and impact on muscle, speculated pathology occurred in the lower leg muscles was stated with providing literature as follows: “Increase of muscle stiffness in active muscle conditions is reported to have several causes, i.e. reflex-mediated stiffness by prolonged muscle contraction after repetitive task which lead to the release of metabolites and activate nociceptors [26, 27], temporary increase of muscle stiffness after repetitive muscle contraction due to static or dynamic exercises, which is reflected by increased intramuscular pressure associated with increases in intramuscular blood flow [28, 29], or static stretch of the muscles by which muscle stiffness increase proportionally to the muscle length [6]. Increase of muscle stiffness in resting muscle conditions is reported to have causes, i.e. increasing collagen mass in muscles by aging and spastic conditions with neuromuscular diseases [7, 30].” “On the other hand, although no measurement by shear wave elastography has been performed, the another suspected causes of increase in muscle stiffness in resting muscles are reported to be fluid retention in the leg due to occupational leg edema after prolonged standing and sitting [12], long-haul flight simulation of sitting 4 to 12 hours [16], and the effect of 4 ours sitting and calf activity [17]. Belczak et al (2018) reported significant increase in leg volume by volumetric measurement using water displacement technique [12]. Mittermayr et al (2007) reported increased lower leg volume by plethysmographic measurement using an optoelectronic scanner system, suggesting the phenomena was due to extravascular fluid shifts in the lower leg. They also measured by ultrasonography the cross-sectional diameter of calf veins, there was no significant change in the diameter by 4 to 12 hours sitting [16]. Singh et al (2017) reported increased leg volume estimated by calf circumference and by bioelectrical impedance analysis. In their study, they suggested two points, first increase in immediate intravascular leg fluid due to hydrostatic pressure and a loss of leg muscle tone, second retention of extravascular fluid in the legs by hydrostatic pressure after 4 hours sitting [17].” 

Response of comment #3) cont’d: Line344, (Line357-359): The following summarized statements was inserted. “Based on literature and result of our study, increase in shear wave velocity with prolonged sitting does not come from change of leg muscle fibers or connective tissue but may come from increase in intra-compartment pressure of the lower leg due to fluid retention in extra-cellular space.” 

Response of comment #4): Line347, (Line371-374): Clinical importance was stated about detection and treatment of leg edema with measurement of shear wave velocity by non-invasive ultrasonography as follows: “The results of this study have clinical importance in the possibility of prevention of muscle stiffness and edema of the lower legs during long time flight or evacuation in the car as refugees after huge earth quake. Measurement of shear wave velocity by non-invasive ultrasonography help people know the extent of leg edema in timely fashion.” 

Methods

The authors report 3ρVs2 because they regarded the muscles as isotropic (Line 190-). However, this is wrong (although the authors are aware).

The orientation of the ultrasound probe is unclear. As far as the reviewer see Figure 4, it seems that the probe was not aligned with the fascicle direction of the muscles. As the authors know, the probe orientation relative to fascicle direction affects the shear wave speed measured, leading to smaller shear wave velocity. In fact, the shear wave velocity reported in this paper is smaller than those in previous studies.

The ICC in this study is smaller than that of previous studies (ICC > 0.9).

The authors stated that no muscle contraction was observed to occur through EMG recording. Please explain in more detail what value the authors determined to be no muscle contractions.

Response of comment #5): Line167-171, (Line170-179): Detail placement of probe and location on the muscle belly was stated in the text as follows: “The location of a probe to be applied on the muscle belly of the medial and lateral gastrocnemius, soleus and tibialis anterior muscles was adjusted in the designated position (the medial and lateral portions of the gastrocnemius muscle at 30 % of the proximal left lower leg, the soleus muscle at 50 % of the left lower leg, and the tibialis anterior muscle at 30 % of the proximal left lower leg [2, 5]) for individual subject. On the most prominent part of the muscle belly, the rectangular shape skin mark with adhesive tape was attached prior to the measurement, and the probe was placed in the rectangular mark (Fig 3a, 3b). Since orientation of muscle fascicles of the gastrocnemius, soleus is oblique by their pennated structure and is redundant due to muscle relaxation, prove setting was aligned to anatomical muscle shortening direction.” 

Response of comment #6): Line203-204, (Line212-213): Statement of EMG activity recording was corrected as follows: “Appearance of EMG activity was determined by observation of action potentials greater than 2SD amplitude of baseline.”

Discussion

The discussion section needs to reflect what the authors found, how it relates to the literature, and then what it means clinically or physiologically. Each paragraph should be logical in sequence as at present; it is a bit hard to follow. Further, the discussion is qualitative but not quantitative.

Response of comment #7): Line329-, (Line332-359): We added in Discussion hypothesized mechanism with two points of view, i.e. one is change of leg muscle fibers and the other is change of extra-cellular space in the compartment. 

Conclusion

Intramuscular pressure was not directly measured in this paper. Thus, the current conclusion is not based on the data of the present study.

Response of comment #8): Line384-391, (Line403-407): Conclusion was revised based on the result of increase in shear wave velocity and speculated pathology in the lower leg muscle as follows: however, in second revision, based on comment from Reviewer #3, Conclusion was finally revised as follows: 

“Shear wave velocity of the lower leg muscles increased with time in 2 hours sitting, and decreased with subsequent leg elevation. Based on the report that the change in the shear wave velocity was proportional to the internal pressure of the leg muscle compartment in turkey models, it is estimated that increase in shear wave velocity with prolonged sitting may come from increase in intra-compartment pressure of the lower leg due to fluid retention in extra-cellular space of the compartment.” 

Response of comment #9): Line43-44, (Line44-45): Conclusion in Abstract was revised based on Conclusion in the Text as follows: “and it is assumed that the increase of shear wave velocity in the 2-hour seated leg is due to fluid retention in extra-cellular space of the compartment.”

Reviewer #3: in first revision 

This study brings descriptive data to the literature on the effect of prolonged sitting on lower limb muscle shear wave velocity (i.e., muscle stiffness under known hypotheses of localized homogeneous, isotropic and purely linear elastic property, which oversimplify the muscle rheology but is of common use in the field). If the angle of insonification with respect to the muscle long axis is defined, this can bring informative data to the literature. The study protocol is quite simple and the sample size is limited in term of number and population characteristics; only normal young male volunteers were scanned. EMG verification of the absence of muscle contraction, which is an important confounder, was made to improve robustness of reported results. Overall, even though it is mainly a descriptive study with no clear rationale or hypothesis, reported results might be of value to design future studies on pathological cohorts of patients with musculoskeletal disorders. A strong effort in restructuring the paper is required, many sections are not at the right place into the manuscript and repetitions are noticed. Please use standard scientific report guidelines.

1) By considering the limited sample size, I would add “a proof-of-concept study” at the end of the title.

Response of comment #1): Line1, (Line2): “a proof-of-concept study” was added in the title as suggested.

2) Abstract, L38-39: There is repetitive information on statistics.

Response of comment #2): Line30-39, (Line39-40): The repetitive part was removed and data in parenthesis was corrected.

3) Introduction, L76: The concept of anisotropic modulus is confusing. L73 is giving the Young’s modulus estimate for a linear, isotropic and purely elastic material, whereas the equation of L76 is simply the shear modulus under same hypotheses. There is no anisotropic modulus leading to this simplified equation, as far as I know.

4) L87: The concept of compartmental pressure increase is indeed of interest to interpret the dataset as the boundary condition imposed by the internal pressure on muscles changes the non-linear strain-stress relation of most biological tissues. Any pre-compressed tissue will result in a displacement of the measured elasticity on the non-linear strain-stress curve and thus will lead to a different estimate of the Young’s modulus given by the equation on L73. This is a limitation of shear wave elastography imaging as linear elasticity is assumed. It would be important to bring basic tissue mechanics concepts into your discussion to avoid misleading interpretation.

Response of comment #3 and 4): Line75-76, (Line72-79): The sentence “On the contrary, if the density of muscles is not uniform, it is expressed in terms of the anisotropic modulus (anisotropic E= ρVs2) [2,7].” was removed to avoid confusion. And the paragraph was revised as follows, “Regarding the difference between elastic modulus and shear wave velocity, Eby (2013) found that the equation E=3ρVs2 (E: elastic modulus kPa, Vs: shear wave velocity m/s, ρ: muscle density 1000 kg/m3) holds if the density of the muscle is assumed to be uniform, in case of measurement along longitudinal muscle fiber direction [4]. On the contrary, if the density of muscles is not uniform, in case of measurement along transverse or oblique muscle fiber direction, it is expressed in terms of the anisotropic modulus (E=ρVs2) [2,7]. In researches of the shear wave velocity in the leg muscles at a resting state (ankle position around the slack angle without muscle contraction), the muscle tissue was measured regarding as the isotropic modulus (E= 3ρVs2) [2,5,7-9].” 

5) L97: Your hypothesis is empty and not based on strong rationales. This should be improved. Why prolonged sitting would change muscle stiffness?

Response of comment #5): Line97, (Line102-106): The hypothesis was added by providing specific pathology in the lower leg muscles as follows: “We hypothesized that the shear wave velocity of the medial and lateral gastrocnemius, soleus and tibialis anterior muscle would increase with time in the resting leg position after 2 hours of sitting and decrease with subsequent leg elevation. Then we considered the mechanism of the velocity change with two points of view, i.e. one is change of leg muscle fibers and the other is change of extra-cellular space in the compartment.”

6) Table 1 is useless and should be removed.

Response of comment #6): Table 1 was removed and detail statement of this result was provided in the Subjects (Line 114-117). 

7) L159: “precision”.

Response of comment #7): Line159: typo error was corrected

8) L186 and 188: What is a “shear wave verbosity” and what is “a constant condition”?

Response of comment #8): Line186: “verbosity” was corrected for “velocity”, and “a constant” was removed as suggested. 

9) L203: What is the output of the Web-1000 system?

Response of comment #9): Line203, (Line207-209): The sentence was corrected as follows: “The electromyography data was recorded at 2000 Hz and collected to a polygraph system of Web-1000 (Nihon Koden Ltd., Tokyo, Japan), then analyzed using LabChart version 7 (AD Instruments, Tokyo, Japan).”

10) Discussion: Remove all repetitions on reported results, this is useless.

Response of comment #10): L283- 297, (Line288-374): All repetitions on reported results were removed from the Discussion

11) L309 and elsewhere: Correct “verbosity”.

Response of comment #11): Line310: All typo error was corrected.

12) L315: Are you sure about this curious hypothesis? How the hydrostatic pressure applied on muscles can be induced by the venous low pressure system of typically a few mmHg? This sounds curious to me.

Response of comment #12): Line309-320, (Line315-320): These two paragraphs were revised to state the effect of hydrostatic pressure in the low leg with sitting position to avoid confusions as follows: “In sitting position, venous pressure of popliteal vein at the popliteal fossa and pressure of saphenous vein at the medial malleolus were reported to be 23.9 mmHg (3.17 kPa) and 54.2 mmHg (7.2 kPa), respectively [25]. Considering the location of shear wave velocity measurements in the tibialis anterior and gastrocnemius muscles in the sitting position, it can be inferred that hydrostatic pressure is equally applied to the tibialis anterior and gastrocnemius muscles through the leg veins for two hours.”

13) L317: You are overstretching the interpretation of your results since you did not measure the intramuscular pressure; consequently you cannot say that this is “in accordance with”.

Response of comment #13): Line317: This paragraph was removed.

14) L327: The muscle length has certainly nothing to do with shear wave speed; you are likely referring to the muscle stretching.

Response of comment #14): Line327, (Line327): We consider the statement of muscle length in this context was correct. 

15) L328: “nature of muscle”, can you be more descriptive and support your hypothesis based on known histopathology analysis?

Response of comment #15): Line328, (Line328): The phrase “nature of muscle” was replaced by “pennation angle of muscle fibers or muscle fiber length” 

16) L330: Here again, verify your hypothesis about venous pressure and impact on muscle. Swelling of muscle versus direct impact of venous pressure are likely different mechanisms. L332: Even though you are citing the literature, more rationale should be given on how an intravenous pressure of typical < 10 mmHg would compress the muscle enough to change the strain-stress non-linear elasticity behavior.

Response of comment #16): Line329-, (Line332-359): To verify hypothesis about venous pressure and impact on muscle, speculated pathology occurred in the lower leg muscles was stated with providing literature as follows: “Increase of muscle stiffness in active muscle conditions is reported to have several causes, i.e. reflex-mediated stiffness by prolonged muscle contraction after repetitive task which lead to the release of metabolites and activate nociceptors [26, 27], temporary increase of muscle stiffness after repetitive muscle contraction due to static or dynamic exercises, which is reflected by increased intramuscular pressure associated with increases in intramuscular blood flow [28, 29], or static stretch of the muscles by which muscle stiffness increase proportionally to the muscle length [6]. Increase of muscle stiffness in resting muscle conditions is reported to have causes, i.e. increasing collagen mass in muscles by aging and spastic conditions with neuromuscular diseases [7, 30].” 

Response of comment #16) cont’d: “On the other hand, although no measurement by shear wave elastography has been performed, the another suspected causes of increase in muscle stiffness in resting muscles are reported to be fluid retention in the leg due to occupational leg edema after prolonged standing and sitting [12], long-haul flight simulation of sitting 4 to 12 hours [16], and the effect of 4 ours sitting and calf activity [17]. Belczak et al (2018) reported significant increase in leg volume by volumetric measurement using water displacement technique [12]. Mittermayr et al (2007) reported increased lower leg volume by plethysmographic measurement using an optoelectronic scanner system, suggesting the phenomena was due to extravascular fluid shifts in the lower leg. They also measured by ultrasonography the cross-sectional diameter of calf veins, there was no significant change in the diameter by 4 to 12 hours sitting [16]. Singh et al (2017) reported increased leg volume estimated by calf circumference and by bioelectrical impedance analysis. In their study, they suggested two points, first increase in immediate intravascular leg fluid due to hydrostatic pressure and a loss of leg muscle tone, second retention of extravascular fluid in the legs by hydrostatic pressure after 4 hours sitting [17]” 

Response of comment #16) cont’d: Line344, (Line357-359): The following summarized statements was inserted. “Based on literature and result of our study, increase in shear wave velocity with prolonged sitting does not come from change of leg muscle fibers or connective tissue but may come from increase in intra-compartment pressure of the lower leg due to fluid retention in extra-cellular space.” 

17) L342: Indeed assessing the venous system with Doppler flow and B-mode diameter measurements would have helped supporting your hypothesis; otherwise your study is mainly descriptive. An impedance plethysmography system would also help assessing electrolyte volume (i.e., blood volume and interstitial inter-compartment water).

Response of comment #17): Line342, (Line346-355): Providing literature, statement of plethysmographic measurement and bioelectrical impedance analysis for leg edema was added in Discussion. 

18) Conclusion: Here again you are overstretching the interpretation of your results; no internal pressure has been reported.

Response of comment #18): Line384-391, (Line406-411): Conclusion was revised based on the result of increase in shear wave velocity and speculated pathology in the lower leg muscle as follows: “Shear wave velocity of the lower leg muscles increased with time in 2 hours sitting, and decreased with subsequent leg elevation. It has been reported that the change in the shear wave velocity was proportional to the internal pressure of the leg muscle compartment. Increase in shear wave velocity with prolonged sitting may come from increase in intra-compartment pressure of the lower leg due to fluid retention in extra-cellular space of the compartment.”

19) L391: “evacuation in the car”, what do you mean?

Response of comment #19): Line391, (Line371-374): We added statement of evacuation in the car as refugees after huge earth-quake as follows: “The results of this study have clinical importance in the possibility of prevention of muscle stiffness and edema of the lower legs during long time flight or evacuation in the car as refugees after huge earth quake. Measurement of shear wave velocity by non-invasive ultrasonography help people know the extent of leg edema in timely fashion [14, 31]” 

At the time of big earthquake named “The east Japan big earthquake in 2011”, many refugees stayed in their own cars for weeks because of lack in housings. Significant number of evacuated people suffered from deep vein thrombosis. 

Reviewer #3 in second revision

Reviewer #3: Overall, your responses to reviewers were badly formulated and it was very difficult to identify if all comments were properly answered. Each reviewer comment should be followed by a clear section indicating your response (e.g., RESPONSE TO COMMENT #1, RESPONSE TO COMMENT #2, etc ...). Also a different color code should be used for each reviewer to allow facilitating identifying changes made in the text. I could notice again the overstretching of the conclusion: "It has been reported that the change in the shear wave velocity was proportional to the internal pressure of the leg muscle compartment", you did not measure any internal pressure! Some typos were also noticed into changes made in the revised document.

No decision can be made unless a new "responses to reviewers" document is made with clear color-coded changes.

Response of comment #1): Response to reviewer’s comments was clearly indicated in rebuttal letter as suggested.

Response of comment #2): A different color code was used for each reviewer in the text; brown for reviewer #1, green for reviewer #2, blue for reviewer #3. 

Response of comment #3): To avoid over-stretching statement in The conclusion, the sentence "It has been reported that the change in the shear wave velocity was proportional to the internal pressure of the leg muscle compartment.", was revised as follows (Line 407-411): “Based on the report that the change in the shear wave velocity was proportional to the internal pressure of the leg muscle compartment in turkey models, it is estimated that increase in shear wave velocity with prolonged sitting may come from increase in intra-compartment pressure of the lower leg due to fluid retention in extra-cellular space of the compartment.”

Response of comment #4): Typo errors were corrected.

In second revision, we added a sentence at the last paragraph of Discussion. L374, It is as follows: “In such cases, it is important to know that simple elevation of lower leg for 3 minutes may reduce leg edema.”

---

## [Editor Report · Decision Letter 2]

28 Apr 2021

Effect of prolonged sitting immobility on shear wave velocity of the lower leg muscles in healthy adults: a proof-of-concept study

PONE-D-20-36699R2

Dear Dr. Aoki,

We’re pleased to inform you that your manuscript has been judged scientifically suitable for publication and will be formally accepted for publication once it meets all outstanding technical requirements.

Kind regards,

Guy Cloutier, Ph.D.

Academic Editor

PLOS ONE
---

## [Editor Report · Acceptance letter]

30 Apr 2021

PONE-D-20-36699R2 

Effect of prolonged sitting immobility on shear wave velocity of the lower leg muscles in healthy adults: a proof-of-concept study 

Dear Dr. Aoki:

I'm pleased to inform you that your manuscript has been deemed suitable for publication in PLOS ONE. Congratulations! Your manuscript is now with our production department. 

Kind regards, 

on behalf of

Dr. Guy Cloutier 

Academic Editor

PLOS ONE